# Diagnosing sea-surface dimethylsulfide (DMS) concentration from satellite data at global and regional scales

Martí Galí[1], Maurice Levasseur[1], Emmanuel Devred[2], Rafel Simó[3], Marcel Babin[1]

[1] Takuvik Joint International Laboratory (Université Laval – CNRS). Biology Department, Université Laval. 1045 Avenue de la Médecine G1V 0A6 Québec (QC) Canada
[2] Fisheries and Oceans Canada, Bedford Institute of Oceanography, Dartmouth, NS B2Y 4A2, Canada
[3] Institut de Ciències del Mar (ICM-CSIC). Passeig Marítim de la Barceloneta 37-49, 08003 Barcelona, Catalonia, Spain

*Correspondence to*: Martí Galí (marti.gali.tapias@gmail.com)

**Abstract.** The marine biogenic gas dimethylsulfide (DMS) modulates climate by enhancing aerosol light scattering and seeding cloud formation. However, the lack of time- and space-resolved estimates of DMS concentration and emission hampers the assessment of its climatic effects. Here we present $DMS_{SAT}$, a new remote sensing algorithm that relies on macroecological relationships between DMS, its phytoplanktonic precursor dimethylsulfoniopropionate (DMSPt) and

plankton light exposure. In a first step, planktonic DMSPt is estimated from satellite retrieved chlorophyll *a* and the light penetration regime as described in a previous study (Galí et al., 2015). In a second step, DMS is estimated as a function of DMSPt and photosynthetically available radiation (PAR) at the sea surface with an equation of the form: $\log_{10}DMS = \alpha + \beta \log_{10}DMSPt + \gamma\ PAR$. The two-step $DMS_{SAT}$ algorithm is computationally light and can be optimized for global and regional scales. Validation at the global scale indicates that $DMS_{SAT}$ has better skill than previous algorithms and reproduces the main

climatological features of DMS seasonality across contrasting biomes. The main shortcomings of the global-scale optimized algorithm are related to (i) regional biases in remotely sensed chlorophyll (which cause underestimation of DMS in the Southern Ocean) and (ii) the inability to reproduce high DMS/DMSPt ratios in late summer and fall in specific regions (which suggests the need to account for additional DMS drivers). Our work also highlights the shortcomings of interpolated DMS climatologies, caused by sparse and biased in situ sampling. Time series derived from MODIS-Aqua in the subpolar

North Atlantic between 2003 and 2016 show wide interannual variability in the magnitude and timing of the annual DMS peak(s), demonstrating the need to move beyond the classical climatological view. By providing synoptic time series of DMS emission, $DMS_{SAT}$ can leverage atmospheric chemistry and climate models and advance our understanding of plankton-aerosol-cloud interactions in the context of global change.

# 1 Introduction

Ocean-emitted gases and particles control the number, size distribution and composition of aerosols in remote oceanic areas (Brooks and Thornton, 2018). These aerosols scatter sunlight and can act as cloud condensation nuclei that alter the radiative properties of clouds, both microscopic (cloud droplet number concentration and effective radius) and macroscopic (cloud

abundance, albedo and lifetime). Interactions between natural aerosols and clouds are a major source of uncertainty in climate projections, confounding the calculation of natural and anthropogenic radiative forcing and the attribution of anthropogenic climate change (Carslaw et al., 2013). Therefore, there is an urgent need to better understand and model the oceanic sources of aerosols, and to better resolve their variations at relevant spatial and temporal scales, from weekly through seasonal and interannual.

The gas dimethylsulfide (DMS) is produced by marine microbial food webs in the sunlit layer of the ocean. With its emission currently estimated at 28 Tg S y$^{-1}$, it contributes about 70% of natural sulfur emissions to the global atmosphere and a major portion of the marine emission of organic volatiles (Carpenter et al., 2012; Schlesinger and Bernhardt, 2013; Simó, 2011). The cloud-seeding activity of DMS and its potential role in climate regulation were first postulated three decades ago

(Charlson et al., 1987; Shaw, 1983). The so-called CLAW hypothesis (Charlson et al., 1987) proposed that a negative feedback could operate between marine phytoplankton, DMS emission and cloud albedo, potentially regulating the Earth's climate. Posterior research showed that the mechanisms behind the potential loop are far more complex than initially envisaged. This, and the estimated low sensitivity of each step of the feedback to changes in its forcing factors, led Quinn and Bates (2011) to refute the CLAW hypothesis.

Nevertheless, atmospheric studies powered by new analytical techniques (Kulmala et al., 2014) and modeling have shown instances where marine DMS controls ultrafine aerosol particle formation in the Arctic (Leaitch et al., 2013; Park et al., 2017), temperate North Atlantic (Sanchez et al., 2018), Antarctica (Yu and Luo, 2010) and the tropical South Pacific atmospheres (Modini et al., 2009). Moreover, Quinn et al. (2017) recently reported that DMS-derived aerosols dominate

cloud condensation nuclei populations over most of the global ocean. Hence, the influence of DMS on marine stratiform cloud albedo remains in the spotlight (Brooks and Thornton, 2018) and the occurrence of a "seasonal CLAW" in remote marine atmospheres is becoming increasingly conceivable (Levasseur, 2013; Vallina and Simó, 2007a).

DMS is produced by marine microbial food webs through a complex network of biological interactions and chemical

processes (Simó, 2004). Its primary source is the enzymatic cleavage of dimethylsulfoniopropionate (DMSP), a multifunctional osmolyte that accumulates at high (hundred mM) intracellular concentrations in some phytoplankton, especially haptophytes, dinoflagellates and some picoeukaryotes (Stefels et al., 2007). DMSP cleavage is catalyzed by a wide diversity of enzymes, called DMSP lyases, produced by some phytoplankton (Alcolombri et al., 2015) and bacteria

(Curson et al., 2011). Breakage of phytoplankton cells through zooplankton grazing, viral attack and autolysis releases DMSP to the algal boundary layer and the dissolved phase and enhances DMS production (Simó, 2004; Stefels et al., 2007). Another process that contributes to DMS production is the diffusive release of DMS from phytoplankton cells, which proceeds almost instantaneously after intracellular DMSP cleavage by DMSP lyases or by photochemically produced radicals (Lavoie et al., 2015; Mopper et al., 2015). Once in seawater, DMS is removed by biotic and abiotic processes. DMS budgets in the upper mixed layer (UML) indicate that, on average, about 90% of dissolved DMS is consumed by bacterial oxidation and UV-driven photolysis, and only 10% is emitted to the atmosphere through turbulent diffusion (Galí and Simó, 2015).

Seawater DMS concentration controls the emission flux because the oceanic UML is largely supersaturated with respect to the atmosphere. DMS concentration in the UML is regulated by a subtle dynamic equilibrium between production and consumption processes with a characteristic timescale of less than 4 days (Galí and Simó, 2015). Over the seasonal cycle, DMS concentration varies mainly in response to the phenology and ecological succession of microbial species and their interplay with physical forcing factors, particularly solar exposure and nutrient supply, which are in turn regulated by vertical mixing (Galí and Simó, 2015; Lizotte et al., 2012; Simó and Pedrós-Alió, 1999). For instance, diatom blooms, typical of nutrient replete conditions at high latitudes, are characterized by low DMSP concentration per unit biomass and low DMSP-to-DMS conversion yield (Lizotte et al., 2012). The opposite is true for microbial communities typical of stratified, nutrient depleted and highly irradiated surface waters, both at low and high latitudes (Galí and Simó, 2010; Lizotte et al., 2012). Under these conditions, two main factors act synergistically to increase DMS concentration (Galí and Simó, 2015; Vallina et al., 2008): the higher contribution of DMSP-rich species to total phytoplankton biomass (Galí et al., 2015; Stefels et al., 2007); and the higher DMSP-to-DMS conversion yield at the microbial community level, possibly caused by the effects of nutrient and irradiance stress (Galí et al., 2013; Stefels, 2000; Sunda et al., 2002, 2007; Vallina et al., 2008). As a result, similar DMS concentrations may occur in waters that differ by one or two orders of magnitude in phytoplankton biomass (Lizotte et al., 2012), and DMS tends to peak in summer across polar to tropical latitudes, lagging the annual chlorophyll peak by some months in the subtropical gyres. The mismatch between phytoplankton biomass, DMSP and DMS, termed the DMS summer paradox (Simó and Pedrós-Alió, 1999), is an essential feature that biogeochemical models strive to reproduce with mixed success (Le Clainche et al., 2010).

With nearly 50,000 DMS measurements taken between 1972 and 2010, the global sea-surface DMS database (https://saga.pmel.noaa.gov/dms/) is a valuable resource for model development and validation. Gridded monthly climatologies (Kettle et al., 1999; Lana et al., 2011) calculated from this dataset are the standard DMS product used as input to atmospheric chemistry and climate models, hence emphasizing the seasonal climatological view (Mahajan et al., 2015; McCoy et al., 2015). At the other end, the climatic role of DMS is often evaluated through extreme sensitivity tests that examine the response of Earth system models to order-of-magnitude perturbations of DMS emission (e.g., Grandey and

Wang, 2015). In comparison, contemporaneous decadal scale DMS variability has received less attention. This gap can be filled using empirical remote sensing algorithms, a handful of which have been developed since the early 2000s (Tesdal et al., 2016; see also the pioneering works of Jodwalis and Benner, 1995, and Thompson et al. 1990). Interestingly, large discrepancies exist among global DMS fields estimated with interpolated climatologies, empirical algorithms or prognostic

biogeochemical models (Tesdal et al., 2016). Although it is tempting to attribute these discrepancies to the poor skill of the models, they may also arise from issues in the calculation of the climatology.

Here we present $DMS_{SAT}$, a new remote sensing algorithm for DMS that proceeds in two steps: (i) estimation of the concentration of the phytoplanktonic DMS precursor, total dimethylsulfoniopropionate (DMSPt), from remotely sensed

chlorophyll and light penetration, and from climatological mixed layer depth (MLD); (ii) estimation of DMS concentration from DMSPt and solar irradiance. This two-step empirical algorithm reflects, with a simplified formulation, the mechanistic understanding of oceanic sulfur cycling described in the previous paragraphs. The DMSPt sub-algorithm was presented by Galí et al. (2015) and is briefly described in Appendix A. Thus, here we focus on the second step, based on the nonlinear relationship between DMS, DMSPt and photosynthetically available radiation (PAR) at the sea surface. We implement our

algorithm to produce a global DMS climatology, which we compare to the last version of the interpolated DMS climatology (Lana et al., 2011) and to climatologies derived from other remote sensing algorithms that follow similar rationales (Simó and Dachs, 2002; Vallina and Simó, 2007). Finally, we implement our algorithm using 14 years of MODIS-Aqua satellite data in the subtropical and the subpolar North Atlantic and in the Northeast Pacific to illustrate and understand interannual DMS variability.

**2 Methods**

**2.1 Datasets used for algorithm development and validation**

In situ concentrations of DMS and DMSPt (nM) and chlorophyll *a* (Chl, mg m$^{-3}$), accompanied by ancillary data (bottom depth, temperature, salinity, wind speed), were downloaded from the global sea-surface DMS database. The latter was complemented with additional in situ datasets recently obtained by the authors' teams. After quality control, the database had

41304, 3700 and 9182 measurements for in situ DMS, DMSPt and Chl, respectively, with 3637 DMS-DMSPt and 8141 DMS-Chl pairs.The in situ database was extended with geophysical and biogeochemical parameters, including satellite retrievals collocated in time and space ("matchups") and gridded climatological datasets, following Galí et al. (2015) (see below). Detailed information regarding data sources, quality control and processing can be found in the SI and in Tables S1-S3.

We performed satellite matchups using SeaWiFS (1997-2010) and MODIS-Aqua (2003-2012) retrievals of remotely sensed Chl (mg m$^{-3}$), vertical attenuation coefficient at 490 nm (Kd490, m$^{-1}$), particulate inorganic carbon (PIC, mol m$^{-3}$) and daily

photosynthetically available radiation at the sea surface (PAR, mol photons $m^{-2}$ $d^{-1}$). To maximize the amount of available matchups, and after verifying the consistency between the two sensors, we produced merged variables by averaging SeaWiFS and MODIS-Aqua matchups. We employed a hierarchical search procedure whereby the matchup criteria were progressively relaxed from one to eight days and from single-pixel to 5x5 pixel bins (SI section S2). These merged variables are hereafter designated with the SAT subscript (e.g. $Chl_{SAT}$). Daily sea surface temperature ($SST_{SAT}$, °C) from the AVHRR sensors was also matched to the database.

The database was further extended with monthly climatological data: daily PAR from SeaWiFS (1997-2010 average); mixed layer depth (MLD, m) from the monthly MIMOC climatology (Schmidtko et al., 2013); bottom depth from the General Bathymetric Chart of the Oceans (GEBCO08); and sea-surface nitrate and phosphate concentrations (µM) from the World Ocean Atlas 2009 (WOA09). Nutricline depths were calculated from WOA09 vertical profiles as the depth where nitrate and phosphate first exceeded 1 µM and 0.4 µM, respectively. Nutricline depth estimations were robust to changes of ±50% in these concentration thresholds.

The mean daily PAR in the upper mixed layer ($PAR_{MLD}$) was calculated as:

$$PAR_{MLD} = [PAR_{SAT} / (Kd490_{SAT} \, MLD)] \, [1 - exp(Kd490_{SAT} \, MLD)] \qquad eq. \, 1$$

When satellite matchups were not available (before September 1997), we used climatological PAR from SeaWiFS (1997-2010 average) in order to increase the temporal coverage of the $PAR_{SAT}$ and $PAR_{MLD}$ variables. Statistical analyses done with climatological or matchup $PAR_{SAT}$ gave very similar results. This procedure was not followed with other variables (Chl, PIC, Kd490) that show wider interannual variations.

### 2.2 Statistical analyses and data binning schemes

All statistical analyses were conducted using (i) non-binned data; (ii) data binned by month and 5°x5° latitude-longitude bins (*M5x5*); and (iii) data binned by month and the 56 Longhurst biogeochemical provinces (*MLongh*) (Longhurst, 2010). Data binning eliminates low-frequency variation ("noise") below a given space or time scale. MLongh binned data were further aggregated into six biomes: two Polar biomes (Arctic and Antarctic), two mid-latitude Westerlies biomes (Northern and Southern hemispheres) one Trades biome (tropical latitudes), and one global coastal biome (Fig. 1c). Variables with a right-skewed approximate lognormal distribution entered statistical analyses after $log_{10}$ transformation: DMS, DMSPt, DMS/DMSPt ratio, Chl, nitrate and phosphate concentrations. To further account for non-normality, we conducted statistical explorations using both bin means and bin medians.

To develop the DMS algorithm we analyzed the relationship between DMS, the DMS/DMSPt ratio and environmental variables (listed in Table 1). After an exploratory analysis based on the calculation of Pearson correlation coefficients (Table 1), we built several regression models where DMS was estimated as a function of in situ DMSPt concentration and additional variables (Table 2 and S4). We added one variable at a time in order of decreasing data availability, and significant terms were selected using stepwise regression with entrance and removal p-values set at 0.001 and 0.005, respectively. The logic for adding one variable at a time, rather than building a single initial model with all the predictor variables, is that the size (N) of the data subset used for model fitting decreases rapidly when variables with sparse coverage are combined. Each set of initial predictors was tested across the three degrees of data binning described above and three degrees of model complexity: linear without interactions, linear with interactions, and quadratic with interactions. This 3x3-nested structure provided a stringent test for the robustness of a given regression model. Improvements in model performance were assessed based on the increase in adjusted r-square, $R^2_{adj}$, and the decrease in root-mean-square error (RMSE) and the Akaike Information Criterion (AIC).

Regression models were further optimized for global and regional domains using the bootstrap method followed by nonlinear optimization as described in SI section 4. Selected models were then validated using an independent data subset composed of in situ DMS measurements and their satellite matchups (described in section 3.1.3) and evaluated using a wide array of skill metrics (following Galí et al., 2015): $R^2$, RMSE, mean absolute percentage error (MAPE), percentage bias, and the slope of major axis (type II) linear regression between observations and model estimates ($Slope_{MA}$). All analyses were carried out using Matlab R2013b.

**2.3 Algorithm implementation**

The newly developed $DMS_{SAT}$ algorithm (Fig. 2) was implemented to produce (i) a monthly global DMS climatology and (ii) several regional time series with 8-day resolution for the period 2003-2016. Further details and data sources can be found in SI section 5 and Table S2.

Global $DMS_{SAT}$ fields were computed using ocean color data from SeaWiFS (1997-2010 monthly climatology, 1/12° grid), SST from AVHRR and the MIMOC monthly MLD climatology. We used SeaWiFS data to maximize the temporal overlap between the satellite-based $DMS_{SAT}$ climatology and the in situ data used to produce the L11 climatology, which span the period 1972-2010. Note, however, that $DMSPt_{SAT}$ climatologies derived from SeaWiFS and MODIS-Aqua are extremely similar (Galí et al., 2015). We established a reference $DMS_{SAT}$ run where $Chl_{SAT}$ was computed with a band-ratio algorithm (OC4-OCI standard NASA algorithm) and the euphotic layer depth ($Zeu_{SAT}$) was computed as the 1% penetration depth of 490 nm radiation ($Zeu_{SAT} = 4.6/Kd490$). The impact of this choice was evaluated with sensitivity tests where $Chl_{SAT}$ and $Zeu_{SAT}$ were calculated with the semi-analytical algorithms of Garver-Siegel-Maritorena (GSM; Maritorena et al., 2002) and Lee et al.(2007), respectively, which are more appropriate in optically complex waters. Observation gaps caused by low

solar elevation at high latitudes in winter were left blank. Global monthly $DMS_{SAT}$ fields were averaged onto 1° and 5° grids for mapping and comparison to other DMS climatologies: the interpolated L11 climatology (Lana et al., 2011), and the climatologies derived with the empirical algorithms of Simó and Dachs (2002) (SD02) and Vallina and Simó (2007b) (VS07). The procedures used to calculate these datasets are briefly described in section 3.2.

Regional $DMS_{SAT}$ time series between 2003 and 2016 were computed using daily MODIS-Aqua data (4.64 km) combined with the MIMOC MLD climatology. As done for the global implementation, we produced $DMS_{SAT}$ fields using both band-ratio and semi-analytical bio-optical products. We also performed a test comparing $DMSPt_{SAT}$ obtained with the MIMOC MLD climatology vs. model-derived MLD time series, showing little $DMSPt_{SAT}$ sensitivity (Fig. S1). Since non-

climatological satellite data contain gaps caused by cloudiness, we applied a binning and gap-filling procedure to obtain full coverage, such that the final regional time series had a resolution of 8 days and 27.8 km. We produced $DMS_{SAT}$ time series for the Bermuda Atlantic Time Series site (BATS; 31°40'N, 64°10'W) and for the northern hemisphere at latitudes >45°N. The latter dataset was then sampled at selected North Atlantic sites and at the Ocean Station P (OSP) in the NE Pacific (50°N, 145°W). Satellite time series were compared to the L11 climatology and to in situ DMS and DMSPt. These in situ

data, kindly provided by the BATS (Levine et al., 2016) and OSP (https://www.waterproperties.ca/linep/) teams, were not used in algorithm development.

## 3 Results

### 3.1 Development and validation of the DMS sub-algorithm

### 3.1.1 Statistical exploration

We analyzed the correlation between potential predictor variables and $\log_{10}(DMS)$ or $\log_{10}(DMS/DMSPt)$ (Table 1). This analysis systematically showed that (i) DMSPt was the best correlate of DMS (r = 0.46 to 0.65), and (ii) surface $PAR_{SAT}$ or mean PAR in the mixed layer ($PAR_{MLD}$) were the best correlates of the DMS/DMSPt ratio (r = 0.35 to 0.67). These correlation patterns remained across different binning levels, suggesting that DMS can be estimated, to first order, by the concentration of its phytoplanktonic precursor compound and by the PAR-dependent enhancement of DMSPt-to-DMS

conversion. It is also noteworthy that the correlation between day length and the DMS/DMSPt ratio was weak or non-significant. This supports the causal relationship between PAR and the DMS/DMSPt ratio and discards other factors that might follow synchronous seasonal cycles.

Guided by the correlation patterns, we established a base regression model expressed by the equation:

$$\log_{10}DMS = \alpha + \beta \log_{10}DMSPt + \gamma \, PAR \qquad \text{eq. 2}$$

This model explained between 50% and 57% of $\log_{10}$(DMS) variance with an increasing level of data binning, and the corresponding RMSE ranged between 0.35 and 0.21 (Table 2).

We assessed whether the base model could be significantly improved by adding predictor variables and/or increasing model complexity. We started by adding a variable X to a linear model without interactions of the form $\log_{10}$DMS = α + β $\log_{10}$DMSPt + γ PAR + δ X. Variable X was chosen among those showing higher correlations to either DMS or the DMS/DMSPt ratio: SST, nitrate concentration, nitracline depth, salinity, wind speed and $PIC_{SAT}$ (Table 1). Although this analysis led us to discard additional predictor variables, its results are briefly described below and compiled in Table S4 for
the sake of completeness. With non-binned data, all the additional variables entered regression models with significant coefficients, but only salinity, wind speed and $PIC_{SAT}$ produced significant decreases in RMSE and AIC. With MLongh binned data, only SST and $PIC_{SAT}$ entered with significant coefficients. Yet, none of the additional variables improved simultaneously the $R^2$adj, RMSE and AIC skill metrics with respect to the base model. Increasing model complexity through addition of interaction and quadratic terms, or by adding a fourth variable, generally resulted in minor improvements or
erratic changes in model performance (results not shown). Invariably, DMSPt and PAR were the only variables with highly significant coefficients ($p \ll 10^{-10}$) regardless of the binning scheme and the inclusion of additional variables.

As a corollary, the use of $PAR_{MLD}$ instead of $PAR_{SAT}$ slightly degraded model skill (Table S4). Although $PAR_{MLD}$ is a priori a more realistic metric of light exposure, it is possible that the use of climatological MLD degraded the $PAR_{MLD}$ estimates.
Another potential explanation is the episodic nature of oceanic vertical mixing, which requires the distinction between the actively mixing layer —defined by higher turbulence than in the ocean interior— and the mixed layer, here termed MLD — defined by the vertical homogeneity of regular temperature-salinity profiles (Brainerd and Gregg, 1995; Sutherland et al., 2014). This distinction implies that, on occasions, mean light exposure at the sea surface is better approximated by surface $PAR_{SAT}$ than by $PAR_{MLD}$. After these considerations we discarded the use of $PAR_{MLD}$ in our algorithm, and focused on
optimizing eq. 2.

Finally, note that $PAR_{SAT}$ is used in our algorithm both as a direct driver of DMS cycling processes and as a proxy for UVR driven processes (e.g., Archer et al., 2010; Galí et al., 2013; Royer et al., 2016). For obvious astronomic reasons, incident PAR and UVR are strongly correlated. Attenuation of PAR and UVR in the atmosphere, first, and in seawater, afterwards,
also covary (Kirk, 2011), so that plankton UVR exposure is to first order well correlated to PAR.

### 3.1.2 Implications of the model structure

Here we analyze the physical and biogeochemical meaning of eq. 2 coefficients in view of their optimization for diagnostic purposes (3.1.3).

- The intercept ($\alpha$) acts to adjust the magnitude of DMS concentrations by a fixed proportion everywhere; e.g., increasing $\alpha$ by $\log_{10}(1.10)$ would raise diagnosed DMS concentrations by 10% globally.

- The $\log_{10}$DMSPt coefficient ($\beta$) expresses the nonlinear relationship between DMS and DMSPt. Fitted $\beta$ is smaller than 1 regardless of the binning applied. For a constant PAR, this implies that DMS increases more slowly than DMSPt (Fig. 3a) and that the DMS/DMSPt ratio decreases nonlinearly and approaches an horizontal asymptote as DMSPt increases (Fig. 3a). This behavior implicitly represents the change in DMSPt-to-DMS conversion efficiency depending on the biomass and structure of the microbial plankton community.

- The PAR coefficient ($\gamma$) expresses the DMSPt-independent modulation of DMS concentration. Fitted $\gamma$ is larger than 0, meaning that the PAR sensitivity increases exponentially with PAR, regardless of DMSPt (Fig. 3b). Incident PAR is positively correlated to shallow mixing at the global scale, and to seawater transparency at low latitudes (but not at high latitudes). Thus, nonlinear PAR sensitivity possibly embodies the enhancement of sunlight exposure by

shallow mixing and deeper PAR (and UVR) penetration. In summary, $\gamma$ represents stress-driven DMS production.

In biogeochemical terms, eq. 2 implies maximal DMS/DMSPt ratios when/where low DMSPt and high PAR co-occur. In biogeographic terms (Fig. 1), highest DMS/DMSPt ratios are found in oligotrophic areas of the Trades biome, where low DMSPt concentrations prevail (<20 nM). Low DMSPt concentrations are also found in winter at high latitudes in deeply

mixed waters, but the corresponding low irradiance results in DMS/DMSPt <0.05. At the high DMSPt concentrations that occur at high latitudes in summer (>100 nM), the DMS/DMSPt ratio is generally <0.1.

As shown in Table 2, eq. 2 coefficients change systematically as the binning spatial scale increases. To further explore the interrelationship between the model coefficients, we used the bootstrap method to produce $10^5$ sets of regression coefficients

for the MLongh dataset. The scatterplots between $\alpha$, $\beta$ and $\gamma$ resulting from the $10^5$ bootstrapped regressions confirmed that covariation between the coefficients is non-random, such that fitted $\beta$ and $\gamma$ are negatively correlated to $\alpha$ (Fig. S2). These trade-offs should be kept in mind when optimizing our model for global or regional implementation.

### 3.1.3 Optimization and validation

By definition, least squares regression minimizes the RMSE, but it has been shown that regression models derived in this

way do not necessarily have the best skill (Jolliff et al., 2009). Therefore, we devised an alternative nonlinear optimization procedure (SI section 4). To obtain realistic solutions, we constrained the optimized coefficients to the confidence intervals derived from the bootstrapped regressions for the MLongh dataset (Fig. S2). The resulting optimal model (eq. 2f) had higher

DMSPt ($\beta$) and PAR ($\gamma$) coefficients and a smaller intercept ($\alpha$), and moved the modeled DMS concentration closer to the 1:1 agreement line without degrading neither RMSE nor $R^2$ (Table 2).

We validated the different versions of eq. 2 (Table 2) by comparing $DMS_{SAT}$ against in situ DMS using an independent subset of the database. Since the complete DMS algorithm proceeds in two steps (Fig. 2), its validation must take into account uncertainty in variables used as input to the $DMSPt_{SAT}$ sub-algorithm. Galí et al. (2015) showed that, apart from the inherent algorithm uncertainty, most uncertainty in $DMSPt_{SAT}$ (RMSE $\leq$ 0.3 in $log_{10}$ space) results from error in $Chl_{SAT}$. Thus, the validation subset was defined according to three criteria: (i) satellite matchup data used as input to the algorithm ($Chl_{SAT}$, $K_{d,490}$, $PAR_{SAT}$ and $SST_{SAT}$) were available; (ii) in situ DMSPt was not available —thus excluding the data used for model fitting; (iii) in situ DMS and Chl were available. We used in situ Chl concentration to constrain the uncertainty in $Chl_{SAT}$ used as input to the algorithm. Indeed, this procedure progressively reduced the size of the validation subset as the maximum tolerated $Chl_{SAT}$ error decreased. Uncertainty arising from $PAR_{SAT}$ could not be assessed because the current database lacks in situ PAR measurements. Frouin et al. (2003) reported an error of $\pm$15% (<10% for weekly and monthly periods), with negligible bias for $PAR_{SAT}$, suggesting it is a minor source of uncertainty.

Fig. 4a summarizes the validation results for the best-performing regression model (eq. 2e) and its optimized version (eq. 2f). Supporting our assumption, $DMS_{SAT}$ skill metrics improved as the maximum tolerated $Chl_{SAT}$ RMSE decreased from 0.5 to 0.2 (Fig. 4). With $Chl_{SAT}$ RMSE smaller than 0.2, the statistics showed erratic behavior owing to reduced sample size. Other skill metrics (not shown in Fig. 4) showed comparable trends.

The optimized model coefficients (eq. 2f) increased $R^2$ and reduced RMSE with respect to the regression-derived coefficients, achieving a maximal $R^2$ of 0.53 and a minimal RMSE of 0.25 for error-free $Chl_{SAT}$ ($log_{10}$-space non-binned data; Table S5). Corresponding best scores in linear space were $R^2 = 0.24$ and RMSE = 3.4 nM. These linear-space statistics might be interpreted as a sign of poor performance, but it should be noted that they were strongly affected by a small fraction of highly biased estimates. Removing the most biased estimates (8% of points beyond a factor of 3 from real measurements; Fig. 4) increased the linear-space $R^2$ to 0.42–0.53 and decreased the RMSE to 1.8–2.3 nM across the full range of $Chl_{SAT}$ error, with MAPE of 34–41% and relative bias of -1% to 7%. Altogether, these statistics illustrate the good performance of the $DMS_{SAT}$ algorithm and the better robustness of log-space statistics. The global-scale optimized algorithm had a mean-normalized standard deviation of 1.06–1.14 ($log_{10}$ space), meaning that the spread of $DMS_{SAT}$ nearly matched that of in situ DMS concentrations.

Table 3 summarizes a detailed analysis of algorithm bias in different biomes. When $Chl_{SAT}$ error was constrained to RMSE < 0.3, $DMS_{SAT}$ showed a global mean bias of 11% (with a global mean bias of 2% in $Chl_{SAT}$ itself). $DMS_{SAT}$ bias was negligible in the Westerlies biomes and larger than $\pm$10% in the other biomes, where its sign often matched that of $Chl_{SAT}$

bias. When $DMS_{SAT}$ bias was assessed using all available matchups, irrespective of $Chl_{SAT}$ error, we obtained a global bias of -9%. In some biomes, the sign and magnitude of $DMS_{SAT}$ bias changed when $Chl_{SAT}$ error was not constrained, indicating the influence of input satellite data. Assuming that $Chl_{SAT}$ matchups (N ~ 15000) are a random sample of the database between 1997 and 2012 (N ~ 24000), this analysis suggests that global $DMS_{SAT}$ bias likely ranges between -9% and 11%.

## 3.2 Global climatologies

We implemented the global scale optimized algorithm (eq. 2f) using the SeaWiFS climatology. As shown in Fig. 5 and 6, DMS concentrations around ~2.5 nM prevail during the astronomic spring and summer in each hemisphere, decreasing to around 1 nM in fall and <1 nM in winter. The seasonal cycle has wider amplitude at high latitudes and is nearly flat in the tropical oceans (see also Fig. S3). Regional enhancement of DMS concentrations occurs in some coastal and shelf areas, equatorial and eastern boundary upwellings, close to the subtropical front in austral summer (40°S), and in the subpolar North Atlantic in boreal summer (60°N). The global mean area-weighted $DMS_{SAT}$ concentration is 1.63 nM (median and geometric mean of 1.36 nM). This global mean decreases by less than 5% when semi-analytical $Chl_{SAT}$ and $Zeu_{SAT}$ products are used instead of our reference products, but larger deviations occur in the coastal biome, particularly in shallow areas, associated to optically complex waters (Table 4).

### 3.2.1 Comparison to the L11 climatology

The L11 DMS climatology (Lana et al., 2011) was calculated using an objective interpolation procedure similar to that used in prior climatologies (Kettle et al., 1999; Kettle and Andreae, 2000). An initial template, called first-guess field, was obtained by calculating the monthly mean DMS in each Longhurst province. The gaps were filled through temporal interpolation and, in provinces with too few documented months, the seasonal cycle was extrapolated by scaling that of neighbor provinces. Objective interpolation was then applied by searching measurements within a 555 km radius, weighting them inversely to the distance from a given grid point, and the resulting global fields were repeatedly smoothed.

The global mean area-weighted $DMS_{L11}$ concentration is 2.43 nM (median 1.88 nM, geometric mean 1.83 nM). Thus, mean $DMS_{SAT}$ concentration is globally 33% lower than $DMS_{L11}$ (Table 3 and 4). The largest and smallest differences occur in the Southern polar and the Coastal biomes, where $DMS_{SAT}$ is 74% and 8% lower than $DMS_{L11}$, respectively (Table 3).

The disagreement between the $DMS_{L11}$ and $DMS_{SAT}$ climatologies varies depending on the regions and the spatial-temporal scales compared. Despite the general offset, the seasonal latitudinal profiles (zonal means) of $DMS_{L11}$ and $DMS_{SAT}$ have similar shapes, and agree very well in June through August. Comparison by means of Hovmöller diagrams (Fig. 6) shows a remarkable qualitative agreement in their month-latitude patterns, except for the polar austral summer. Fig. 6 also reveals smaller disagreements in the Arctic Ocean in winter-spring and in the equatorial band during most of the year, with lower concentrations in $DMS_{SAT}$ in both cases. The most striking regional disagreements appear when $DMS_{SAT}$ and $DMS_{L11}$ are

compared by means of seasonal anomaly maps (Fig. 5). The sign of the $DMS_{SAT}$-$DMS_{L11}$ divergence changes from positive to negative in a patchy pattern, often following the boundaries of the Longhurst biogeochemical provinces.

### 3.2.2 Comparison to the SD02 climatology

The SD02 algorithm (Simó and Dachs, 2002) was designed to estimate DMS from MLD and $Chl_{SAT}$ using two different
equations depending on the Chl/MLD ratio,

$$DMS = -\ln(MLD) + 5.7 \qquad\qquad Chl/MLD < 0.02 \;\;(\text{eq. 3a})$$
$$DMS = 55.8\; Chl/MLD\; + 0.6 \qquad\qquad Chl/MLD \geq 0.02 \;\;(\text{eq. 3b})$$

such that DMS increases linearly with the Chl/MLD ratio in stratified productive conditions (e.g. high latitudes in summer) and inversely with MLD in typical oligotrophic conditions.

Validation of SD02 with the same dataset used for $DMS_{SAT}$ indicates that it explains less variance ($\log_{10} R^2$ of 0.22–0.31) but has similar RMSE and MAPE (Table S5). Globally, $DMS_{SD02}$ is characterized by a bimodal distribution (Fig. 7), with an
area-weighted mean of 2.12 nM, 13% lower than $DMS_{L11}$ (Table 4). SD02 estimates are in good agreement with the L11 climatology at tropical and temperate latitudes. In the Southern Westerlies biome, however, prevailing deep vertical mixing and low Chl cause SD02 to underestimate DMS throughout the productive season (Figs. 6 and S3). Another feature of SD02 is the high DMS concentration in Northern polar latitudes through late summer and fall, caused mainly by the shallow MLD due to freshwater-driven stratification. As $DMS_{SAT}$, $DMS_{SD02}$ suffers a negative bias in the Antarctic biome during the
productive season (November through February).

### 3.2.3 Comparison to the VS07 climatology

Vallina and Simó (2007b) reported a globally valid linear relationship between DMS concentration and the solar radiation dose (SRD) in the upper mixed layer in the global ocean, according to the equation:

$$DMS = 0.492 + 0.019\; SRD \qquad\qquad (\text{eq. 4})$$

This numerical relationship was not meant to be used as a diagnostic algorithm but as an evidence for the emerging response of ecosystem DMS production to changes in solar radiation. However, the fact that SRD explained large part of the variance of DMS concentration across regions and seasons prompted its use in global warming projections (Vallina et al., 2007). SRD
is analogous to $PAR_{MLD}$ (eq. 1), but replacing $PAR_{SAT}$ by total shortwave irradiance ($Ed_{SW}$; W m$^{-2}$). Here we implemented VS07 with two variations: (i) we used $Kd490_{SAT}$ instead of a fixed Kd (note that in phytoplankton-rich and continentally-influenced waters, $Kd490_{SAT}$ is generally higher than the fixed Kd = 0.06 m$^{-1}$ used by Vallina and Simó (2007b)); (ii) we

estimated $Ed_{SW}$ from $PAR_{SAT}$ by converting the latter to units of W m$^{-2}$ (Morel and Smith, 1974) and then applying a constant $Ed_{SW}/PAR_{SAT}$ ratio of 1/0.43 (Kirk, 2011).

VS07 shows poorer performance than $DMS_{SAT}$ and SD02 when validated with individual measurements (Table S5). It produces rather uniform DMS fields compared to the other climatologies (Fig. 6 and 7c), with mean area-weighted concentration of 2.71 nM (Table 4). VS07 performs well in the Westerlies biome, especially in the northern hemisphere, but invariably overestimates (underestimates) DMS in the Trades (Polar) biomes (Fig. S3).

### 3.3 Regional $DMS_{SAT}$ time series

We selected different regions to test the $DMS_{SAT}$ algorithm based on the following criteria: the abundance of in situ data (subpolar North Atlantic), the existence of seasonal and multiannual time series (Ocean Station P and Bermuda Atlantic Time Series), and the challenges posed by intra- and interannual variability of phytoplankton and DMS(P) in each region.

### 3.3.1 Subpolar Atlantic and Pacific

We used MODIS-Aqua data to produce a 14-year $DMS_{SAT}$ time series (and the corresponding climatology) for the northern hemisphere at latitudes >45° N. In this regional implementation we used a different set of coefficients, obtained from regression of M5x5 binned data restricted to latitudes >45° N (eq. 2g, Table 2). These regional coefficients largely corrected the negative bias observed with globally optimized coefficients. In this case, further optimization did not lead to significant improvement. We then sampled the resulting time series in some representative regions: three rectangles with an area of ~200,000 km$^2$ each, located along the 50°N–56°N band in the North Atlantic, and the Ocean Station P (OSP, 50° N, 145° W) in the NE Pacific.

Fig. 8 shows the seasonal cycles of $DMS_{SAT}$, $DMSPt_{SAT}$ and $Chl_{SAT}$ in selected North Atlantic areas: (a) the deep waters of the northwest Atlantic drift, (b) the shelf break west of Ireland, and (c) the shallow Southern North Sea. We observe a good agreement between the 14-year $DMS_{SAT}$ climatology and the L11 climatology, except in the Southern North Sea where $DMS_{SAT}$ is too high through summer and fall. $DMS_{SAT}$ reproduces well the east-west variation in the temporal lag between the annual peaks of DMS and Chl, with a lag of up to four months in the Southern North Sea. The most salient result is however the wide interannual variability of the $DMS_{SAT}$ seasonal cycles. Diagnosed $DMS_{SAT}$ concentrations during the productive season vary by up to threefold between years (see variability metrics in Fig. 8), and the annual $DMS_{SAT}$ peak can occur within a temporal window of 2–3 months. Although years with a major peak in spring-summer are the norm, a second peak in late summer is not unusual. Additional validation supports the good performance of $DMS_{SAT}$ in the subpolar North Atlantic (Fig. S4), lending credit to satellite-diagnosed variability patterns.

We used the same MODIS-Aqua dataset to analyze the mean seasonal cycle and the interannual variability at Ocean Station P (Fig. 9a-c), where DMS has been measured around February, June and August since 1996. $DMS_{SAT}$ captures in situ DMS concentrations in February and June but suffers a low bias in August. Examination of August measurements during the 2005-2016 period suggests the existence of two regimes: 8 years have in situ DMS of $6.6 \pm 1.1$ nM, about twice as high as $DMS_{SAT}$, and 4 years have much higher in situ DMS of $16.1 \pm 4.8$ nM, about six fold higher than $DMS_{SAT}$. Local tuning of eq. 2 using OSP data could not increase $DMS_{SAT}$ in August-September without degrading its performance in other months.

Finally, note that these time series were calculated using semi-analytical bio-optical products (see section S5). Using the band-ratio Chl algorithm for MODIS (OC3) gave very similar results in deep ocean regions but 70% higher concentrations in the shallow Southern North Sea, possibly due to interference of non-algal materials on OC3 Chl retrieval (data not shown).

### 3.3.2 Bermuda Atlantic Time Series

Using the globally tuned coefficients (eq. 2f), $DMS_{SAT}$ reproduces the shape of the mean seasonal cycle at the oligotrophic BATS station but underestimates DMS by around twofold between June and October (Fig. 9d-e). In August, part of this bias can be attributed to $DMSPt_{SAT}$ (Fig. 9f). However, replacing $DMSPt_{SAT}$ by in situ DMSPt raises $DMS_{SAT}$ by only 17%, indicating that most of the underestimation is caused by the DMS sub-algorithm. Optimizing the coefficients using local data (eq. 2h; section S4) improves the model-data fit by decreasing the DMSPt coefficient and increasing the PAR coefficient. Indeed, different studies have shown that irradiance suffices to explain most of the DMS seasonal cycle at BATS (Galí and Simó, 2015; Toole and Siegel, 2004; Vallina and Simó, 2007b). Regarding interannual variation, it is of note that the locally tuned $DMS_{SAT}$ is in excellent agreement with in situ data throughout 2007 and in June through August in all years, while the underestimation persists in September and October of 2006 and 2008.

### 4 Discussion

The $DMS_{SAT}$ algorithm captures in situ variability (Fig. 4–9 and S3) using a small set of predictor variables (Fig. 2). Moreover, it reproduces the mismatch between DMS and Chl such that, at a given $Chl_{SAT}$ concentration, diagnosed DMS can vary by up to 40-fold. This mismatch is larger than that produced by the SD02 or the VS07 algorithms, but still smaller than in the database (Fig. 7d-h). The progressive dissociation between $Chl_{SAT}$ and DMS imposed by the two-step algorithm, and the nonlinear relationships embodied in eq. 2 (Fig. 3), allow $DMS_{SAT}$ to produce a DMS peak in the right concentration range in summer across different latitudes, despite order-of-magnitude variations in chlorophyll concentration (Le Clainche et al., 2010).

Independent validation using satellite matchups suggests that $DMS_{SAT}$ estimates are globally within ±10% of in situ measurements. However, this is at odds with the global mean bias of -33% suggested by comparison to the L11 gridded climatology. In some regions, assessments of $DMS_{SAT}$ bias based on comparison to matchups and to L11 are consistent

(Table 3), indicating shortcomings in the algorithm and/or in input satellite data. For instance, the globally optimized coefficients produce too low DMS in northern high latitudes, which we solved through regional tuning, building on the relative abundance of DMS(P) measurements north of 45°N (Table 2, Fig. 8). In southern high latitudes, too-low $DMS_{SAT}$ primarily results from the negative $Chl_{SAT}$ bias (estimated at <-50% by Johnson et al. (2013) and -56% in our matchup dataset). Improving $DMS_{SAT}$ in this region would require, in first place, the improvement of bio-optical algorithms and an important sampling effort to better document DMS(P) dynamics (Fig. 1; Jarníková and Tortell, 2016). In contrast to high latitudes, database matchups in the Westerlies and Trades biomes suggest smaller $DMS_{SAT}$ bias than comparison to L11 gridded data (Table 3). While some of this bias certainly arises from too-low $DMS_{SAT}$ in late summer at specific locations (Fig. 9), it is also plausible that divergences between $DMS_{SAT}$ and $DMS_{L11}$ arise from regional biases in the interpolated climatology.

In what follows we pinpoint the strengths and weaknesses of our novel approach focusing on two aspects. In section 4.1 we examine how geo-statistical shortcomings of the in situ DMS database cause bias in the L11 interpolated climatology, highlighting the advantages of $DMS_{SAT}$ and paving the way towards improving gridded DMS fields. In section 4.2 we speculate about potential causes behind the occurrence of high DMS/DMSPt ratios in late summer and early fall. This exercise shows our limited capacity to account for relevant biogeochemical processes and explain their interannual variation using satellite data, and identifies knowledge gaps that need to be tackled to improve diagnostic and prognostic modelling of oceanic DMS(P). The rationale is that both kinds of issues ("geo-statistical" and "biogeochemical") are highlighted by discrepancies among in situ data, L11, and the macroecological relationships embodied in our algorithm.

## 4.1 Known sources of error and bias: interpolated climatology versus $DMS_{SAT}$

Global DMS fields estimated by the L11 climatology and by the $DMS_{SAT}$ algorithm show remarkable geographic differences (Fig. 5), and their mean concentrations differ by a factor of ~1.5. Particularly, changes in the sign of the $DMS_{SAT}$–$DMS_{L11}$ anomaly often follow the somewhat artificial boundaries of the Longhurst biogeochemical provinces. As explained below, regional and global biases in L11 arise from the application of objective interpolation procedures to a global dataset characterized by: (1) the right-skewed distribution of DMS concentrations (Kettle et al., 1999; Fig. 7), (2) the small amount of monthly data available in many biogeochemical provinces (Fig. 1), (3) the absence of repeat measurements in most oceanic regions, (4) the low spatial resolution of most DMS datasets, and (5) the preferential sampling of DMS-productive conditions.

A primary drawback for the calculation of interpolated climatologies is the overrepresentation of biologically productive conditions in the sea-surface DMS database. This sampling bias is clearly illustrated by comparing SeaWiFS-retrieved Chl concentration in the global ocean and in the database matchups (Fig. 7a). If we assume that SeaWiFS matchups (N = 11600) represent a random sample of the DMS database (N = 41304), and considering the global positive correlation between Chl

and DMS (Fig. 7d), this implies a sampling bias towards high DMS concentrations. The bias is largest when the comparison is restricted to the spring-summer semester of each hemisphere, when the median $Chl_{SAT}$ is 0.19 in the global SeaWiFS climatology and 0.74 in the DMS database matchups. This is the period when DMS peaks and has more influence on mean annual DMS concentration.

Sampling bias is intertwined with the right-skewed statistical distribution of sea-surface DMS concentrations (Fig. 7b) and the poor spatial resolution of most in situ DMS datasets. Spatial averaging, justified by data scarcity, is appropriate when applied over small or sufficiently homogeneous regions. However, when applied over a large and potentially heterogeneous Longhurst province, it can propagate the sampling bias over the entire province and up to the global scale. As illustrated in

Fig. 6g, the mean DMS concentration in M5x5 bins is systematically higher, by 40% on average, than the corresponding median. Province-level averaging converts the long tail of high in situ DMS concentrations into too-large province means, such that the global mode of $DMS_{L11}$ is similar or even higher than that of in situ DMS (Fig. 7b).

The influence of extreme in situ DMS concentrations is maximal in productive regions, where mean/median ratios of around

4 are observed in M5x5 bins (Fig. 6g). In these regions, sharp productivity gradients and dynamic ecosystem processes complicate the task of sampling DMS through all appropriate scales (Nemcek et al., 2008), suggesting the need to apply finer-scale or dynamic regionalization (Devred et al., 2007) prior to interpolation. Emerging biogeochemical relationships, like that between net community production and DMS observed by Kameyama et al. (2013) in the northeast Pacific, might assist DMS interpolation and diagnosis, but require validation across contrasting regions and relevant scales (Asher et al.,

2017). In low latitude oligotrophic areas where DMS shows reduced spatial variability (Royer et al., 2015), the method used to construct interpolation-based climatologies (Kettle et al., 1999a; Lana et al., 2011) seems appropriate regarding spatial resolution.

In the temporal domain, the calculation of interpolated climatologies is complicated by two factors: poor interannual

coverage, i.e. the scarcity of DMS measurements repeated in different years in a given region; and poor seasonal coverage, i.e. the scarcity of fully resolved seasonal cycles. Seasonal coverage is limited at the province level (see Table 1 in Lana et al., 2011) and obviously worse in 5-degree bins (Fig. 1d). Regarding interannual coverage at the MLongh level, 42% of the province-month bins contain measurements from a single year, 21% from two years, and 37% from three or more years. Thus, data from one or two years are often assumed representative of the mean ecosystem state in interpolated climatologies,

which is probably not the case in regions with wide interannual variability or long-term trends (Vantrepotte and Mélin, 2011). While this does not necessarily bias global DMS fields, it can produce artificial seasonal cycles. For example, L11 suggests the existence of early spring and fall DMS peaks in the North Atlantic drift area, which result from interpolation from neighbor regions (Fig. 8a). In contrast, $DMS_{SAT}$ suggests these are improbable (spring) or infrequent (fall) features. Another example is found at OSP, where $DMS_{L11}$, based on measurements made before 2003, is in poor agreement with

measurements made between 2005 and 2016. In February and June, $DMS_{SAT}$ is in better accordance with in situ DMS data at OSP (Fig. 9a-b).

In summary, caution has to be taken when comparing DMS measurements, their derived climatological products, and
independent model estimates that are not collocated in time. This temporal mismatch may partly explain the poor correlation between modeled DMS climatologies, on one hand, and the DMS database and $DMS_{L11}$ climatology, on the other (Tesdal et al., 2016). Note that the latter study compared DMS fields binned into monthly 5°x5° boxes (M5x5), such that 82% of the bins contained measurements from a single year.

Compared to interpolated climatologies, $DMS_{SAT}$ provides a robust means to estimate DMS concentrations in sparsely sampled areas because it relies on satellite observations and macroecological relationships. The resulting DMS fields are in better accordance with natural gradients in plankton abundance (biogeography, phenology) and environmental forcing, *as long as the models can account for the driving factors*, as discussed below below.

### 4.2 Unknown sources of error: How far can we go with remote sensing algorithms?

By testing $DMS_{SAT}$ in challenging biogeochemical settings (Fig. 9), we identified its main drawback: the failure to reproduce high DMS, and more specifically DMS/DMSPt ratios higher than 0.3, at intermediate PAR levels (Fig. 3), as observed between midsummer and early fall at BATS and OSP *in some years*. This limitation can hardly be fixed without identifying the underlying biogeochemical processes, which are not necessarily the same in these contrasting biogeochemical regimes. Although this feature is probably not widespread (see figure 2 in Lana et al., 2011), its occurrence in emblematic time series
stations warrants further discussion.

A common explanation could be the underestimation of irradiance effects, caused by the use of sea-surface $PAR_{SAT}$ rather than $PAR_{MLD}$ as DMS predictor variable. For instance, a delay of seasonal mixing, associated with deeper irradiance penetration, could enhance stress driven DMS production well into fall. Yet, examination of the BATS and OSP time series
does not support this explanation. At both sites, the summer MLD is stable at about ≤20 m and deepens slowly in late summer (Levine et al., 2016; Steiner et al., 2012), such that $PAR_{MLD}$ declines faster than sea-surface $PAR_{SAT}$ (eq. 1). Thus, using $PAR_{MLD}$ instead of surface PAR cannot delay the decline of modeled DMS/DMSPt ratios through the summer, and other factors need to be invoked.

In the oligotrophic BATS station, some modeling studies proposed macronutrient limitation of bacteria (Polimene et al., 2011) and also phytoplankton (Belviso et al., 2012) as drivers of the seasonal mismatch between DMSPt and DMS, besides irradiance (Vallina et al., 2008). With this in mind, we tried to factor phosphate and nitrate limitation into our regression models using different variables: nutrient concentrations, nutricline depths (Table S4) and limitation factors estimated

according to Michaelis Menten kinetics (not shown). However, none of the tested variables improved the regression models significantly. Moreover, macronutrient availability (limitation) terms generally entered regression models with positive (negative) coefficients, even when regressions were restricted to oligotrophic low latitudes. This implies that macronutrient limitation of phytoplankton growth globally acts to decrease DMS, offsetting nutrient stress responses that increase DMS.

Note also the irregular occurrence of high DMS at BATS in late summer in different years (Fig. 9d; Levine et al., 2016),and that a BATS-like seasonality is not observed at other sites with late summer macronutrient limitation (Archer et al., 2009; Belviso et al., 2012; Galí and Simó, 2015; Vila-Costa et al., 2008). Altogether, these observations suggest that regional macronutrient stress responses are difficult to generalize.

Analysis of the OSP time series also yields valuable information because macronutrient concentrations remain at high concentrations in late summer in this iron-limited regime (Harrison et al., 2004). While in situ DMS is accurately estimated by $DMS_{SAT}$ in February and June, the variable DMS peak occurring around August is strongly underestimated. Previous studies emphasized the role of iron limitation at OSP, which configures phytoplankton communities dominated by high-DMSP taxa (Asher et al., 2017; Levasseur et al., 2006; Royer et al., 2010; Steiner et al., 2012). Interestingly, $DMSPt_{SAT}$

peaks in August at OSP, in phase with the in situ DMS peak and in good accordance with the few available DMSPt measurements (Fig. 9c), even though the $DMSPt_{SAT}$ sub-algorithm does not explicitly resolve phytoplankton taxonomy (Galí et al., 2015). Therefore, high DMS yields, possibly co-occurring with low DMS removal rate constants (Asher et al., 2017), are required to explain DMS/DMSPt ratios observed at OSP. High DMS yields probably result from a combination of processes, including algal and bacterial DMSP metabolism (Merzouk et al., 2006; Royer et al., 2010) and microzooplankton

grazing (Steiner et al., 2012). The striking late summer variability at OSP is presently not captured by biogeochemical models (Steiner et al., 2012) or empirical algorithms, and it remains unanswered whether it simply reflects too low sampling frequency, or it is caused by processes that switch on/off depending on environmental conditions on a given year, or by the variable location of oceanic fronts in response to circulation patterns.

In summary, our analysis indicates that additional factors are needed to better reproduce DMS seasonality in specific regions, where DMS/DMSPt ratios are occasionally higher than the "baseline" established by eq. 2 (Fig. 3 and 9). Biotic interactions involving phytoplankton, bacteria and microzooplankton, regionally interacting with iron and macronutrient limitation in multiple ways, are good candidates to explain strong deviations from the mean relationship between DMS, DMSPt and irradiance. However, they can hardly be represented in empirical algorithms with our current level of

understanding, particularly when interannual changes are considered (Fig. 8 and 9).

From a practical standpoint, tuning the eq. 2 coefficients is a workable alternative in certain regions (Fig. 8 and 9). If sufficient measurements were available in all oceanic areas, eq. 2 could perhaps be generalized in a way that allowed its coefficients to vary across different biogeochemical regimes, while avoiding geographic discontinuities. The inclusion of

additional terms in eq. 2 lacks strong statistical support when applied globally, at least with the current dataset (Table S4). If posterior analyses supported the addition of new satellite variables, their retrieval uncertainty and its propagation to $DMS_{SAT}$ should be considered. More obviously, climatological variables such as the WOA nutrient concentrations are not appropriate to produce time series, and their use in remote sensing algorithms should be minimized. The only climatological variable used in our algorithm is MLD, which enters mainly as a categorical variable (Galí et al., 2015), such that $DMSPt_{SAT}$ is robust to MLD uncertainties (Fig. S1).

The question of the "optimal model complexity" is a pervasive one in biogeochemistry, and the right answer may depend on the purpose of each study. The algorithms tested here showed improved qualitative and quantitative performance with increasing complexity (VS07 < SD02 < $DMS_{SAT}$). VS07 failed to capture DMS patterns outside the subtropical band, possibly due to its inability to modulate the DMS-irradiance relationship depending on phytoplankton biomass. Inclusion of phytoplankton biomass-dependent terms in SD02, and of implicit taxonomic information through the embedded $DMSPt_{SAT}$ sub-algorithm in $DMS_{SAT}$, improved algorithm skill in productive regions, where DMS shows wider seasonal cycles and sharper spatial gradients.

More sophisticated approaches may be needed to achieve significant improvements in model skill, but they also suffer from major uncertainties. For example, neural networks were successfully used to estimate DMS in the Arctic (Humphries et al., 2012), but their robustness might be compromised by the small training datasets, the use of climatological variables and the lack of a mechanistic basis. Complex biogeochemical models with satellite data assimilation have strong potential for resolving interannual DMS variations, but reliance on several tens of poorly constrained parameters currently limits their skill (Le Clainche et al., 2010; Galí and Simó, 2015; Tesdal et al., 2016). An approach of intermediate complexity that deserves further exploration is DMS diagnosis based on a simplified steady-state budget equation (Galí and Simó, 2015). Applying empirical parameterizations for the main DMS production and removal pathways, this approach could enhance the flexibility of remote sensing algorithms across a wider range of biogeochemical settings.

## 5 Conclusions and outlook

Sensors on polar-orbiting satellites provide synoptic observations of the global ocean surface every few days, and are thus well suited to resolve spatial and temporal variations in DMS concentration. The $DMS_{SAT}$ algorithm presented here, based on robust macroecological relationships, reproduces the main spatial-temporal features of sea-surface DMS(P) concentrations with remarkable skill using satellite retrieved Chl, euphotic layer depth and PAR and climatological MLD. Other strengths of our approach are its flexibility, allowing for regional tuning, and the minimal computing cost.

When compared against the L11 interpolated DMS climatology (Lana et al., 2011), the $DMS_{SAT}$ climatology shows similar latitudinal profiles but disagrees in the basin-scale patterns. Examination of spatial DMS statistics highlights possible

shortcomings in the L11 climatology caused by the combination of sparse and biased sampling, the right-skewed distribution of DMS, and the interpolation procedures used. High-resolution measurements of DMS(P), if validated against traditional standard techniques (Royer et al., 2014), will help improving interpolated climatologies and models.

The global mean area-weighted $DMS_{SAT}$ concentration is 1.63 nM, 33% lower than $DMS_{L11}$ (2.43 nM). Global-scale $DMS_{SAT}$ fields are insensitive to the choice of different Chl and euphotic depth satellite products, but semi-analytical products should be used in optically complex coastal waters to avoid DMS overestimation(after detailed examination at regional scale). Globally, $DMS_{SAT}$ suffers a negative bias for exogenous and endogenous reasons. In the Antarctic Ocean, it is affected by the negative bias in satellite-retrieved chlorophyll. At the BATS and OSP stations (at least), additional factors

besides irradiance are needed to enhance DMS concentration in late summer in some years. Excluding the Coastal and Antarctic biomes, $DMS_{SAT}$ bias assessed using satellite matchups ranges between -16% and -20% (Table 3). This bias is probably more realistic than the -33% deduced from comparison to L11, given the evidence for DMS overestimation in the L11 climatology.

Gauging global DMS emission is critical to understand gas-to-particle conversion efficiency and the dynamics of CCN populations in the marine boundary layer. Assuming a linear relationship between global mean DMS concentration and emission (figure 8 in Tesdal et al., 2016), $DMS_{SAT}$ suggests a global emission of 16–20 Tg S $y^{-1}$ (depending on the assigned bias). These emission values lie within the low range of current estimates (Lana et al., 2011; Tesdal et al., 2016).

Unlike climatologies constructed from the database, the satellite-based algorithm allows to explore interannual change. Implementation of $DMS_{SAT}$ in the subpolar Atlantic between 2003 and 2016 illustrates the wide interannual variability in the timing and magnitude of the annual DMS peak(s) over large areas. This opens new avenues for studying the imprint of oceanic aerosol precursors on cloud properties using simultaneous ocean-atmosphere satellite observations (Krüger and Grabßl, 2011; McCoy et al., 2015; Meskhidze and Nenes, 2006). If coupled to atmospheric measurements and numerical

models, $DMS_{SAT}$ enables studying the effects of contemporaneous DMS variability on atmospheric chemistry and clouds, which could lead to a better understanding of intricate aerosol-cloud interactions. Further work is warranted to analyze marine DMS emission variability patterns in regions where climate is particularly sensitive to DMS, such as the Southern Ocean and the Arctic.

**Data availability**

The primary database used to develop the $DMS_{SAT}$ algorithm is publicly available at http://saga.pmel.noaa.gov/dms/. The database extended with satellite matchups and climatological variables can be provided by the authors on request, as well as the global DMS and DMSPt climatologies derived with the $DMS_{SAT}$, SD02 and VS07 algorithms. The L11 DMS

climatology and other related documents and datasets can be downloaded from https://www.bodc.ac.uk/solas_integration/implementation_products/group1/dms/.

**Code availability**

The code used to perform the data analyses and produce $DMS_{SAT}$, SD02 and VS07 DMS fields can be provided by the
authors on request.

**Supplements**

A supplementary information file is available.

**Author contributions**

M.G. designed the study, performed the research and wrote the paper, with input from all coauthors through the different
phases. E.D. processed remote sensing reflectance data used as input for the northern hemisphere $DMS_{SAT}$ time series.

**Competing interests**

The authors declare that they have no conflict of interest.

**Acknowledgments**

We thank the NASA Ocean Biology Distributed Active Archive Center (OB.DAAC) for access to MODIS and SeaWiFS
datasets, T.S Bates (NOAA/ PMEL) for the maintenance of the GSS DMS(P) database and the DMS-GO project for the recent database update (a joint initiative of the SOLAS Integration Project and the EU projects COST Action 735 and EUR-OCEANS to R.S.); Eric Rehm and Maxime Benoît-Gagné for IT support; Naomi Levine, John Dacey, Nick Bates and Scott Doney for sharing BATS data; and Marie Robert and Michael Arychuk for guidance on access to Ocean Station P public datasets. We acknowledge funding from the Canada Excellence Research Chair in Remote Sensing of Canada's New Arctic
Frontier (M.B.), the Canada Research Chair on Ocean Biogeochemistry and Climate and a NSERC Discovery Grant Program and Northern Research Supplement Program (M.L.), the NETCARE network (funded under the NSERC Climate Change and Atmospheric Research program) and ArcticNet (The Network of Centres of Excellence of Canada). R.S. acknowledges funding from the Spanish Ministry of Economy through project BIOGAPS. M.G. acknowledges the receipt of a Beatriu de Pinós postdoctoral fellowship funded by AGAUR (Generalitat de Catalunya). This project is a contribution to
the research program of Québec-Océan and the Takuvik Joint International Laboratory (CNRS-France & Univeristé Laval-Canada).

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

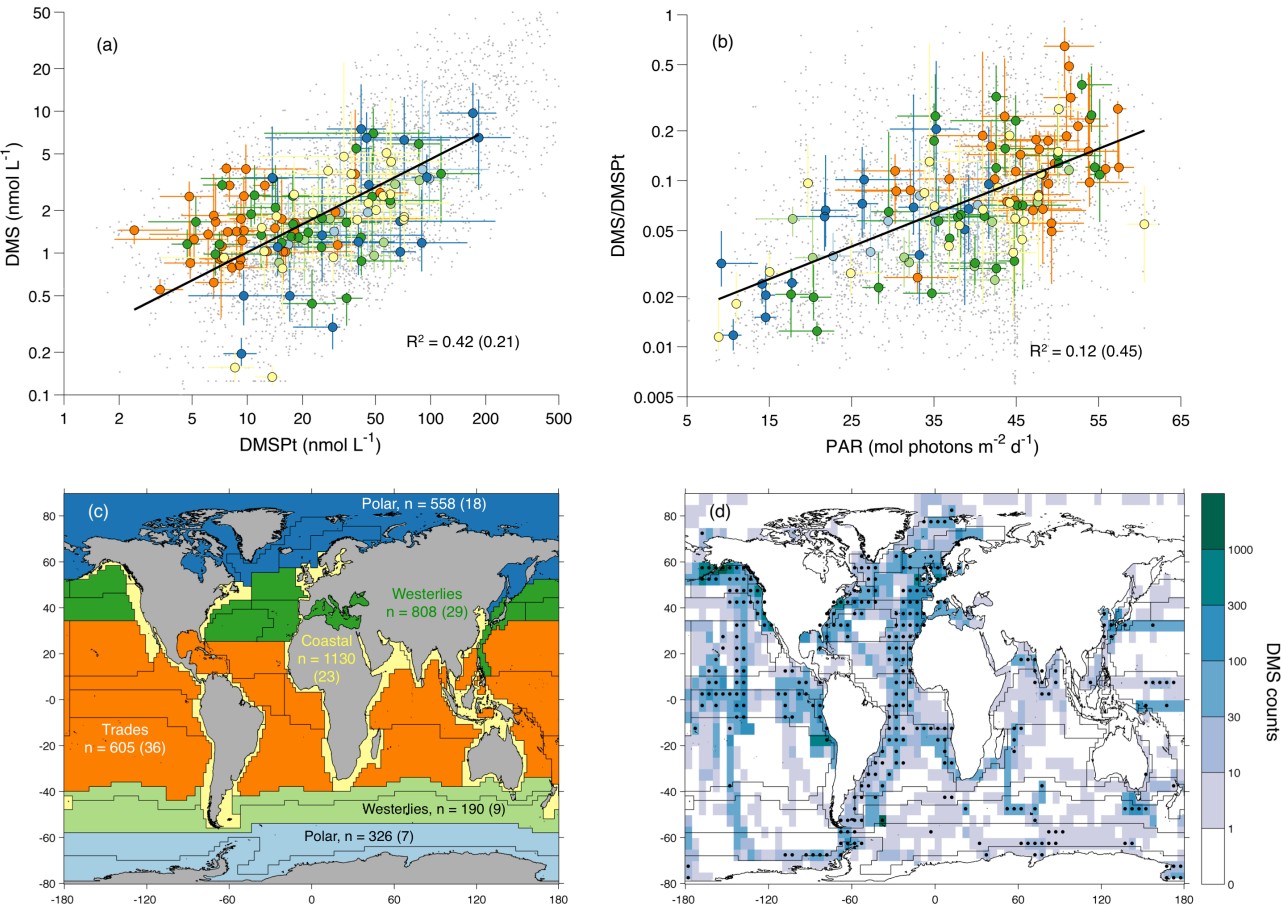

**Figure 1: Relationship between DMS, DMSPt and PAR across oceanic biomes and data availability**. (a) DMS versus DMSPt. (b) DMS/DMSPt ratio versus mean daily irradiance (PAR) at the sea surface. (c) Longhurst biogeochemical provinces and biomes. (d) In situ DMS data counts in 5°x5° latitude-longitude bins; stippling indicates bins with measurements available in 3 or more months. In (a) and (b) small grey dots represent individual data points and large colored dots represent the median in a given Longhurst biogeochemical province and month and the corresponding interquartile ranges. Province-month medians are colored by biome following the map in (c), which also shows the amount of DMS-DMSPt-PAR measurements available in each biome. The $R^2$ and data counts outside and inside parentheses correspond to non-binned data and province-month (MLongh) binned data, respectively. Regression lines in (a) and (b), calculated with MLongh binned data, are only illustrative.

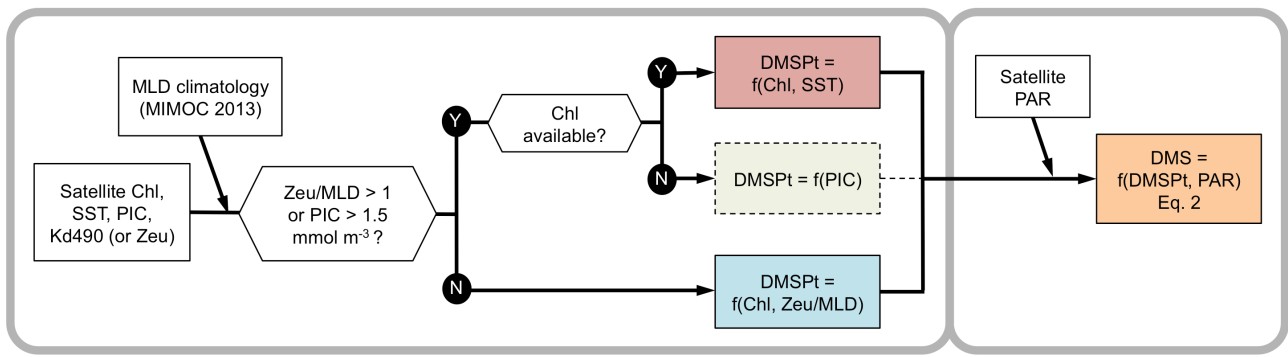

**Figure 2: Scheme of the DMS$_{SAT}$ algorithm**. The algorithm proceeds in two steps: the DMSPt sub-algorithm (described by Galí et al., 2015 see Appendix A) and the DMS sub-algorithm (this study). Dashed lines mark the PIC-based equation of the DMSPt sub-algorithm, which in practice is not used when gap-free satellite Chl fields are used as input.

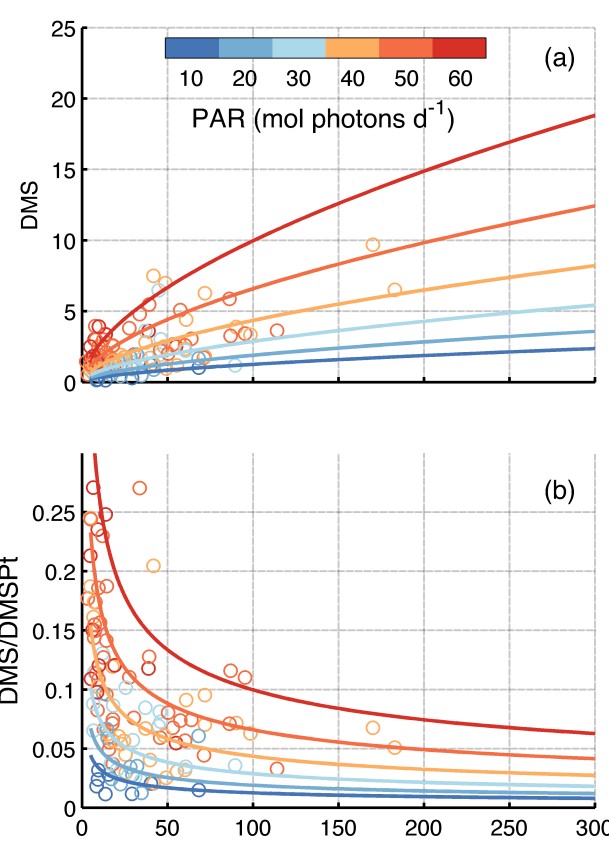

DMS versus [DMSPt,PAR] optimized fit (eq. 2f)

$$\log_{10}\text{DMS} = -1.237 + 0.578\log_{10}\text{DMSPt} + 0.0180\text{PAR}$$

**Figure 3: Relationship between DMS, DMSPt and PAR as represented in the DMS$_{SAT}$ algorithm**. (a) DMS vs. DMSPt and (b) DMS/DMSPt ratio as a function of PAR. Colored circles represent the medians of in situ data binned by Longhurst province and month, and lines correspond to the model predictions. PAR levels are indicated in the color bar.

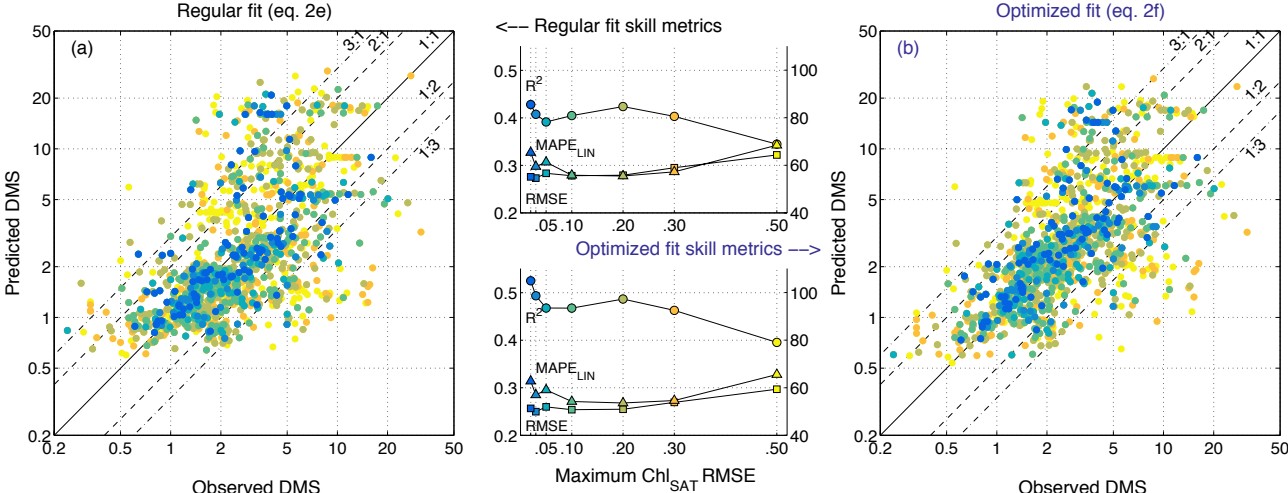

**Figure 4: Algorithm validation results constrained by the uncertainty in satellite-retrieved Chl**. (a) eq. 2e, derived from regular multiple regression; (b) eq. 2f, obtained through an optimization procedure. The scatterplots compare non-binned data and model predictions, color-coded depending on the maximum tolerated error in $Chl_{SAT}$ with respect to Chl in situ, as shown in the x-axis of the center plots. The center plots show the performance of the DMS algorithm for increasing error in $Chl_{SAT}$, evaluated with different skill metrics: the $\log_{10}$ space $R^2$ and RMSE (left y-axis) and the linear space MAPE (right y-axis). N increases from 86 to 1293 as the tolerated $Chl_{SAT}$ error increases.

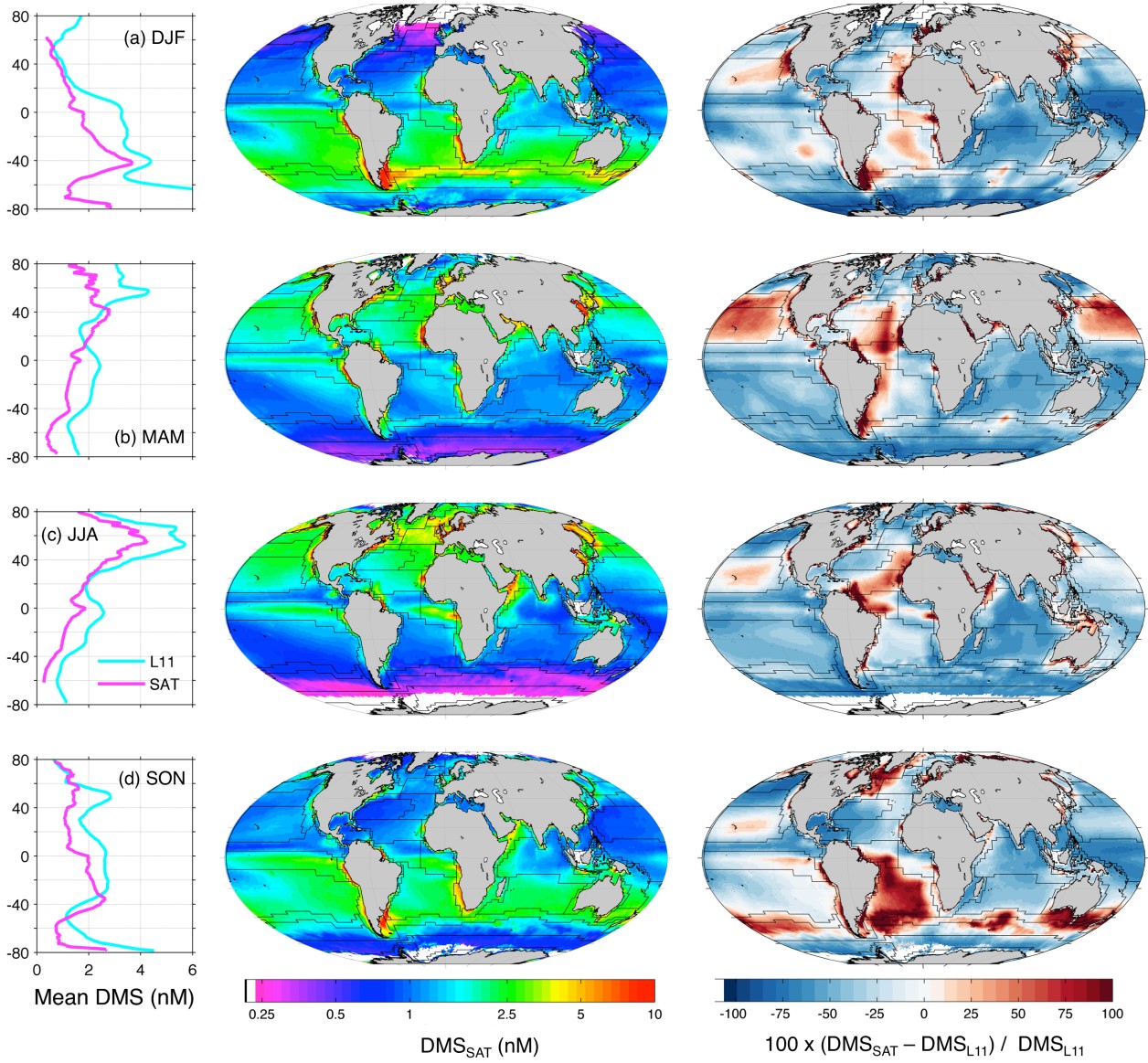

**Figure 5: Global DMS$_{SAT}$ concentration fields by season**. (a) December-February (DJF); (b) March-May (MAM); (c) June-August (JJA); (d) September-November (SON). Each row contains mean latitudinal profiles for the L11 climatology and DMS$_{SAT}$ (left); DMS$_{SAT}$ concentration maps (center); and maps of the % difference between DMS$_{SAT}$ and the L11 climatology (right).

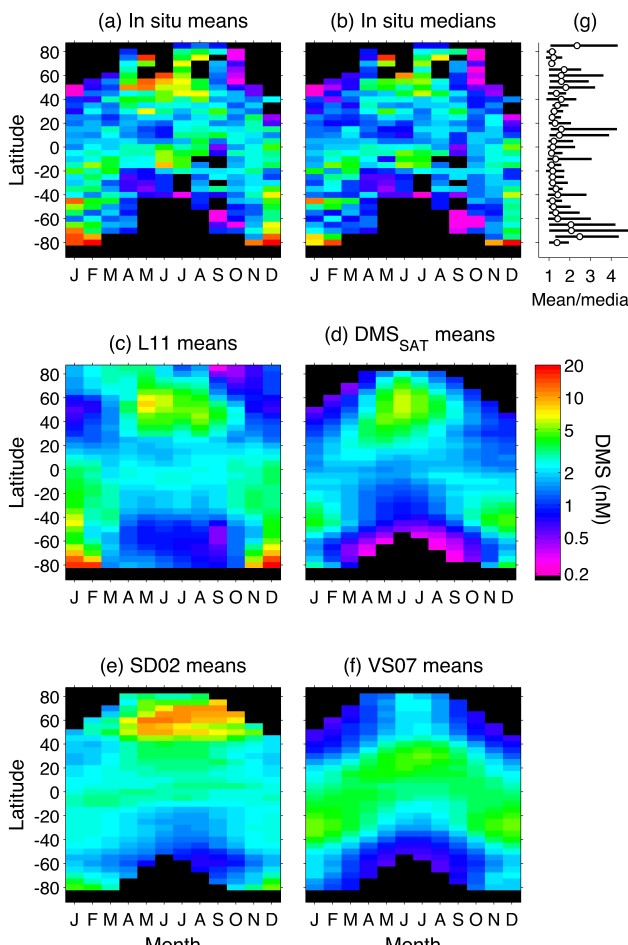

**Figure 6: Hovmöller diagrams comparing climatological DMS fields (monthly 5-degree latitude bins)**. (a) in situ database means, (b) in situ database medians, (c) L11 climatology, (d) DMS$_{SAT}$ algorithm, (e) SD02 algorithm, and (f) VS07 algorithm. Panels (a-f) share the same color scale; the black background indicates no data. Panel (g) shows the average and range of monthly mean/median ratios (shown in panels a and b) in each latitude bin. Geometric means of in situ binned data (b) are similar to bin medians.

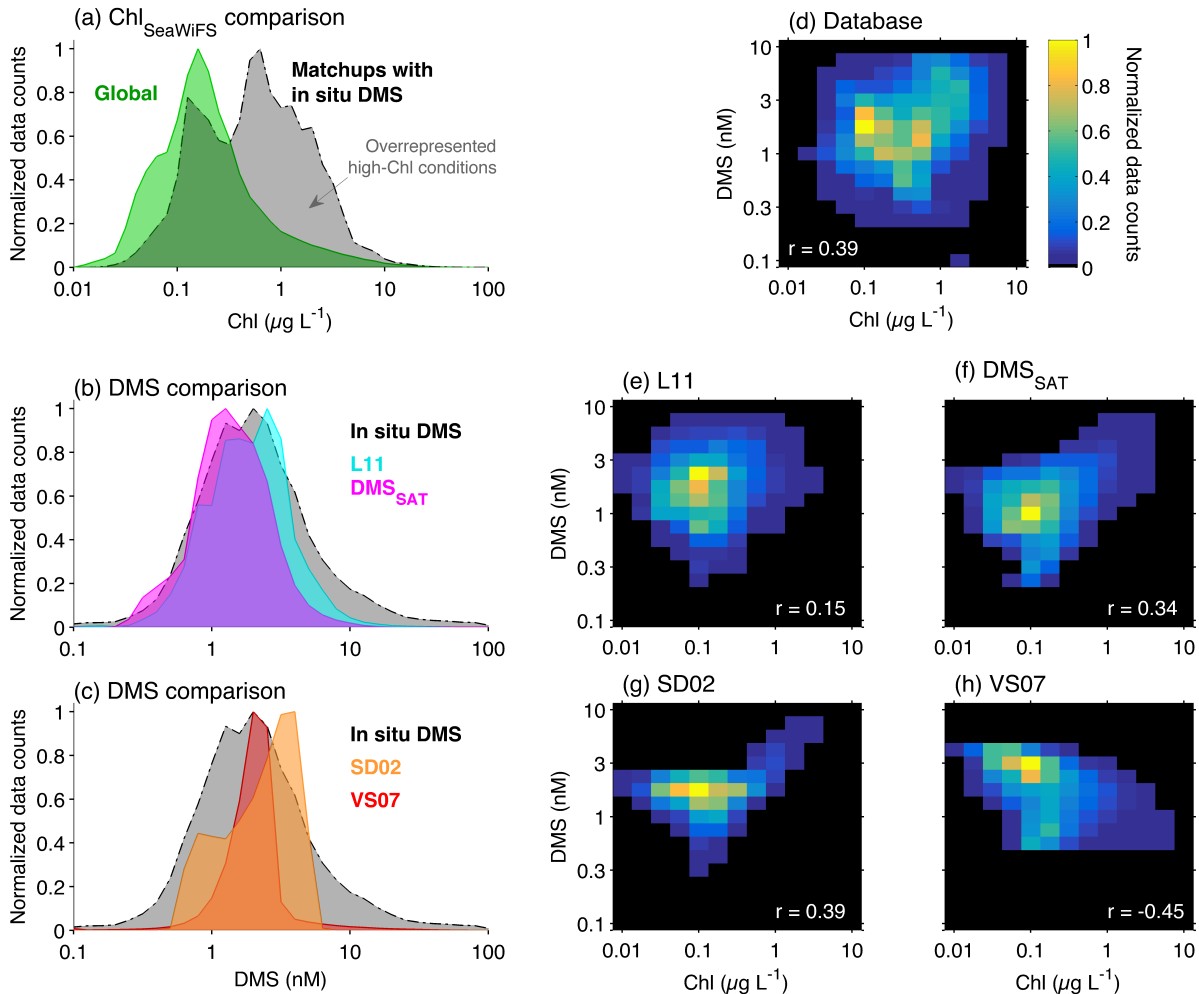

**Figure 7: Histograms illustrating the relationship between DMS and Chl in the global ocean**. (a) Global SeaWiFS 1997-2010 $Chl_{SAT}$ climatology and SeaWiFS $Chl_{SAT}$ matchups for the in situ DMS database; (b) Global L11 and $DMS_{SAT}$ climatologies and in situ DMS database; (c) Global SD02 and VS07 climatologies and in situ DMS database; (d) bivariate histogram of in situ DMS versus corresponding SeaWiFS $Chl_{SAT}$ matchups; (e–h) bivariate histograms of the global SeaWiFS $Chl_{SAT}$ climatology versus climatological DMS from (e) L11, (d) $DMS_{SAT}$, (e) SD02 and (f) VS07 algorithms. In (d-h), the Pearson correlation ("r") between DMS and Chl in each dataset is super-imprinted in white.

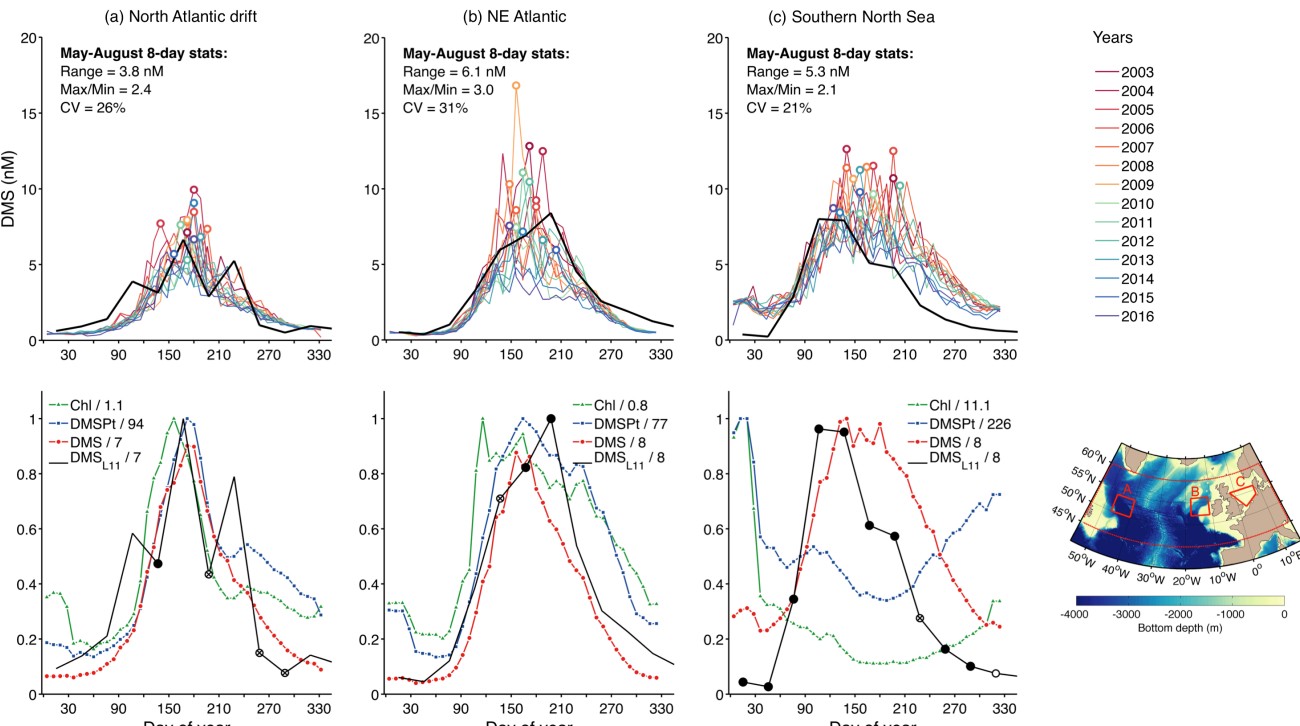

**Figure 8: Interannual DMS$_{SAT}$ variability in the subpolar North Atlantic**. (a) Northwest Atlantic drift, (b) shelf break west of Ireland, (c) Southern North Sea shelf. Top panels show individual years between 2003 and 2016 diagnosed from 8-day MODIS-Aqua data, marked by colors, and the mean seasonal cycle according to the L11 DMS climatology (black); colored circles mark the peak of each seasonal cycle. Bottom panels show the mean annual cycles of Chl$_{SAT}$, DMSPt$_{SAT}$, DMS$_{SAT}$ and the L11 DMS climatology; each variable is divided by its maximum, shown by the number in the quotient; a common scaling factor is used for DMS$_{SAT}$ and DMS$_{L11}$; markers on the L11 line indicate the amount of in situ data on which the L11 climatology is based in a given month: no data, i.e. month filled through interpolation (no marker); 1–9 measurements in one single year (empty circles); ≥10 measurements in one single year (crossed circles); and ≥10 measurements distributed in different years (filled black circles). Red polygons on the map show the 3 selected areas and the larger region used for the validation scatterplots (see Fig. S4).

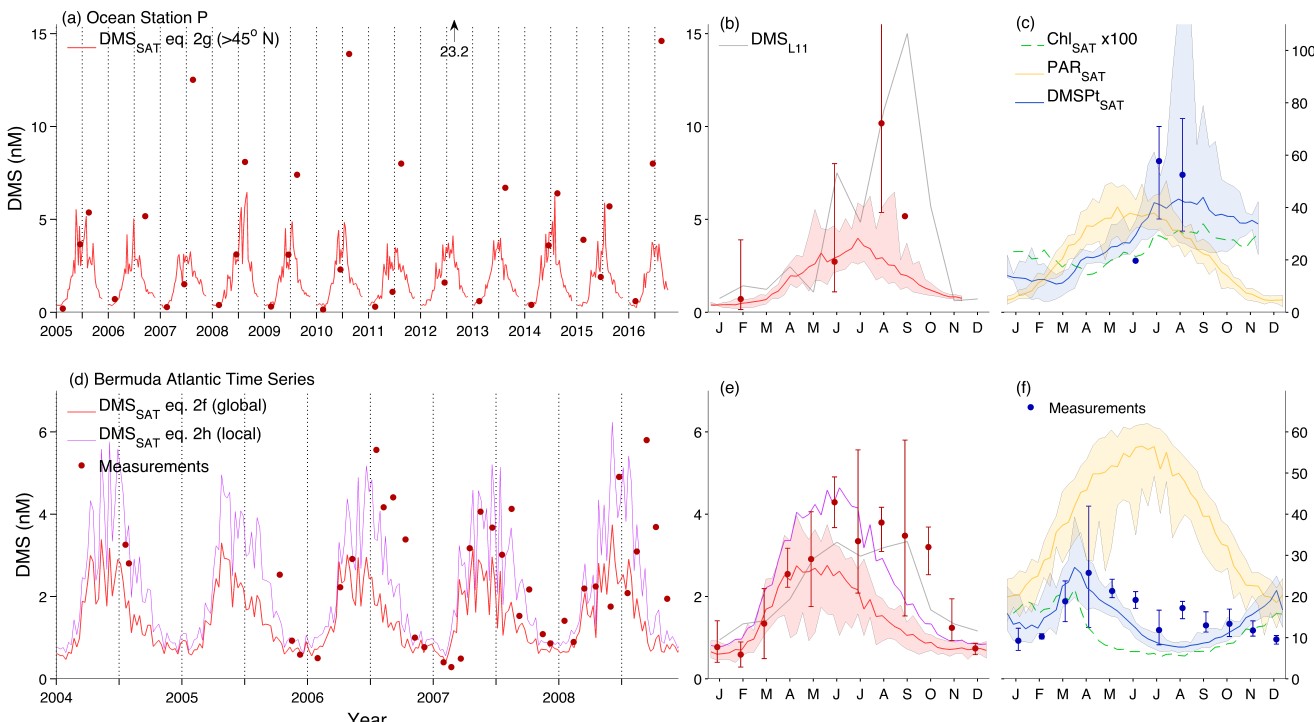

**Figure 9: DMS$_{SAT}$ *vs*. in situ data at long-term research stations**. (a–c) Ocean Station P in the northeast Pacific (50°N, 145°W); (d–f) Bermuda Atlantic Time Series station (31°40'N, 64°10'W). (a) and (d) compare DMS$_{SAT}$ estimates to in situ

5   measurements; (b) and (e) compare monthly DMS means (i.e. climatologies) derived from DMS$_{SAT}$ (2003-2016 MODIS-Aqua data), in situ data (available measurements between 2003-2016) and the L11 climatology; (c) and (f) show the corresponding Chl$_{SAT}$, DMSPt$_{SAT}$ and PAR$_{SAT}$ climatologies (2003-2016) and in situ DMSPt. Lines and shaded envelopes show the mean ± range of satellite-derived data; filled circles and error bars show the mean ± range of available in situ data. DMS and DMSPt are in nM units, Chl in μg L$^{-1}$, PAR in mol photons m$^{-2}$ d$^{-1}$. In addition to public OSP data (three DMS

10   measurements per year generally available between 2005 and 2016) we show DMSPt data reported in published studies (Asher et al., 2017; Levasseur et al., 2006; Royer et al., 2010). BATS data comprise the monthly 2005-2008 time series (Levine et al., 2016) and cruise data from the vicinity of BATS in July 2004 (Bailey et al., 2008; Gabric et al., 2008).

**Table 1: Correlation analysis for different data binning levels**. Correlation coefficients (r) with p-value < 0.01 are not shown; *italic* marks 0.0001 < p < 0.01; na: not applicable. "Ratio" refers to $\log_{10}(DMS/DMSPt)$. DMS, DMSPt, Chl, MLD, $[NO_3]$ and $[PO_4]$ were $\log_{10}$ transformed; Pearson's r calculated on $\log_{10}$-transformed variables were higher than those calculated on the same non-transformed variables, and similar in magnitude to Spearman's rank correlations. Bottom and nutricline depths have positive sign (deeper is bigger). See the text for other acronyms.

| | Non-binned data | | | | Monthly 5°x5° binning (M5x5) | | | | Monthly Longhurst binning (MLongh) | | | |
|---|---|---|---|---|---|---|---|---|---|---|---|---|
| | **DMS** | N | **Ratio** | N | **DMS** | N | **Ratio** | N | **DMS** | N | **Ratio** | N |
| **In situ data** | | | | | | | | | | | | |
| DMSPt | 0.65 | 3637 | na | 1442 | 0.58 | 308 | na | 157 | 0.46 | 122 | na | 87 |
| Chl | 0.45 | to | -0.33 | to | 0.37 | to | -0.28 | to | 0.34 | to | -0.45 | to |
| SST | -0.02 | 41304 | 0.29 | 3637 | | 1562 | 0.45 | 308 | | 322 | 0.56 | 119 |
| Salinity | -0.12 | | 0.27 | | | | | | | | 0.32 | |
| Wind speed | -0.12 | | -0.13 | | -0.12 | | | | *-0.20* | | *-0.27* | |
| Bottom depth | -0.19 | | 0.10 | | -0.12 | | *0.16* | | | | *0.26* | |
| Day Length | 0.42 | | 0.06 | | 0.43 | | | | 0.49 | | | |
| **Climatological data** | | | | | | | | | | | | |
| MLD | -0.37 | 35505 | -0.13 | 3433 | -0.32 | 1474 | | 298 | -0.51 | 312 | *-0.24* | 119 |
| $[NO_3]$ | 0.06 | to | -0.19 | to | 0.16 | to | -0.31 | to | | to | -0.45 | |
| $[PO_4]$ | 0.05 | 39478 | -0.15 | 3637 | 0.13 | 1535 | -0.34 | 308 | | 318 | -0.32 | |
| N-cline | -0.14 | | 0.30 | | -0.22 | | 0.44 | | | | 0.55 | |
| P-cline | -0.12 | | 0.24 | | -0.14 | | 0.45 | | | | *0.37* | |
| **Satellite match-up data** | | | | | | | | | | | | |
| PAR$_{SAT}$ | 0.32 | 16411 | 0.35 | 1123 | 0.30 | 498 | 0.46 | 124 | 0.52 | 171 | 0.67 | 86 |
| PAR$_{MLD}$ | 0.12 | to | 0.37 | to | 0.15 | to | 0.49 | to | 0.36 | to | 0.66 | to |
| Chl$_{SAT}$ | 0.37 | 41088 | -0.42 | 3620 | 0.22 | 1539 | -0.34 | 307 | *0.28* | 321 | *-0.39* | 119 |
| PIC$_{SAT}$ | 0.24 | | -0.27 | | 0.29 | | *-0.30* | | 0.33 | | | |

**Table 2: Summary of fitted model coefficients and goodness-of-fit statistics**. Different sets of coefficients were obtained by fitting the model $\log_{10}DMS = \alpha + \beta \log_{10}DMSPt + \gamma PAR$ to observed DMS, DMSPt and $PAR_{SAT}$ after applying different binning schemes. Equations 2f and 2h were derived using a different optimization procedure, applied to the global MLongh binned dataset and to the Bermuda Atlantic Time Series local dataset. Shading highlights the models implemented to calculate a global DMS climatology (lighter gray) and regional or local time series (darker gray).

| | Binning | Bin metric | Equation | $\alpha$ | $\beta$ | $\gamma$ | $R^2_{adj}$ | RMSE | SlopeMA | N |
|---|---|---|---|---|---|---|---|---|---|---|
| **Regression, global scale** | Non-binned | | 2a | -1.213 ± 0.028 | 0.672 ± 0.012 | 0.0136 ± 0.0006 | 0.50 | 0.35 | 0.62 | 3620 |
| | M5x5 | Mean | 2b | -1.154 ± 0.083 | 0.669 ± 0.0371 | 0.0130 ± 0.0015 | 0.55 | 0.28 | 0.67 | 307 |
| | | Median | 2c | -1.061 ± 0.084 | 0.569 ± 0.039 | 0.0130 ± 0.0015 | 0.46 | 0.28 | 0.58 | |
| | MLongh | Mean | 2d | -1.061 ± 0.115 | 0.583 ± 0.054 | 0.0155 ± 0.0019 | 0.57 | 0.24 | 0.70 | 118 |
| | | Median | 2e | -1.018 ± 0.100 | 0.452 ± 0.050 | 0.0163 ± 0.0016 | 0.57 | 0.21 | 0.69 | |
| **Optimization, global scale** | MLongh | Median | 2f | -1.237 | 0.578 | 0.0180 | 0.56 | 0.22 | 0.87 | 118 |
| **Regression, regional scale (>45N)** | M5x5 | Mean | 2g | -1.283 ± 0.154 | 0.670 ± 0.097 | 0.0186 ± 0.011 | 0.68 | 0.28 | 0.80 | 87 |
| **Optimization, local scale (BATS)** | Non-binned | | 2h | -0.898 | 0.316 | 0.0214 | 0.44 | 0.26 | 0.66 | 35 |

**Table 3**. Relative deviation ("bias") of the global-scale optimized DMS$_{SAT}$ algorithm across ocean biomes when evaluated against the in situ database (a and b) and against the L11 monthly climatology (c). Comparison with the in situ database was done using either (a) satellite matchups with Chl$_{SAT}$ error constrained with respect to in situ Chl or (b) all Chl$_{SAT}$ matchups, regardless of the availability of in situ Chl. The last column shows a qualitative overall assessment of DMS$_{SAT}$ bias, based on columns a-c, classifying the most likely magnitude and sign as small± (±10%), moderate (10 to 40%, either + or -) or large (>40%, either + or -).

| Biome | In situ database matchups | | | | | c) L11 gridded data | | DMS$_{SAT}$ bias assessment |
| | a) Chl$_{SAT}$ RMSE < 0.3 | | | b) All matchups | | | | |
| | Bias DMS$_{SAT}$ | Bias Chl$_{SAT}$ | N[a] | Bias DMS$_{SAT}$ | N[a] | Bias DMS$_{SAT}$ | Area[b] | |
|---|---|---|---|---|---|---|---|---|
| Polar N | -28% | -7% | 54 | -39% | 2291 | -33% | 3.0% | Moderate- |
| Westerlies N | -4% | -11% | 125 | -13% | 1872 | -32% | 10.1% | Small± |
| Trades | -14% | 7% | 442 | 6% | 3489 | -34% | 56.6% | Small± to moderate- |
| Westerlies S[c] | -3% | -30% | 30 | -27% | 1585 | -25% | 14.2% | Small± to moderate- |
| Polar S | -59% | -6% | 42 | -47% | 170 | -74% | 5.7% | Large- |
| Coastal | 46% | 4% | 377 | 3% | 5673 | -8% | 10.4% | Small± to large+? |
| Global ocean | 11% | 2% | 1053 | -9% | 15080 | -33% | 100% | Small± to moderate- |
| Global ocean -pS -C | -16% | -2% | 634 | -20% | 9249 | -30% | 83.9% | Moderate- |

[a]For the database comparisons, the amount of data (N) available for validation in a given region depends on the total amount of measurements, the proportion of data points with satellite matchups and, for the constrained case, the fraction of matchups where Chl$_{SAT}$ has log$_{10}$ RMSE < 0.3 compared to concurrent Chl in situ. Samples with available DMSPt measurements were excluded because they were used in model fitting and optimization.

[b]For the global climatology comparison we report the % of ocean area, excluding pixels that could not be observed by satellites (high latitude winter).

[c]Samples where DMSPt was measured not excluded for this biome due to the small amount of data. Removing them would leave N = 13.

**Table 4: Mean area-weighted DMS concentrations for different climatological datasets and domains**. Different results for the DMS$_{SAT}$ algorithm were obtained using alternative approaches for retrieving chlorophyll $a$ concentration (Chl$_{SAT}$) and the euphotic layer depth (Zeu$_{SAT}$). Results (based on 1°x1° gridded data) are shown for the global domain, the Longhurst coastal biome and, within it, areas shallower than 200 m (shelves); na: not applicable.

| DMS algorithm or data product | Chl$_{SAT}$ product | Kd$_{SAT}$ or Zeu$_{SAT}$ product | Area weighted DMS mean (nM) | | |
|---|---|---|---|---|---|
| | | | Global | Coastal biome[a] | Coastal biome shelves[a] |
| L11 climatology (Lana et al., 2011) | na | na | 2.43 | 2.70 | 2.56 |
| SD02 (Simó and Dachs, 2002) | OC4-CI | na | 2.12 | 3.07 | 4.27 |
| VS07 (Vallina and Simó, 2007b) | na | Kd490 | 2.71 | 2.70 | 2.48 |
| DMS$_{SAT}$ (eq. 2f and Appendix A, this study) | OC4-CI | Zeu = 4.6/Kd490 | 1.63 | 2.49 | 3.06 |
| | GSM | Zeu = 4.6/Kd490 | 1.58 | 2.42 | 3.09 |
| | | Zeu_Lee | 1.55 | 2.41 | 3.02 |

[a]The Longhurst coastal biome represents about 10.4% of the global ocean area, of which one third is shallower than 200 m (3.5% of global area).

**Appendix A. Description of the DMSPt$_{SAT}$ sub-algorithm**

We estimated sea-surface DMSPt concentration (nmol L$^{-1}$) using the algorithm of Galí et al. (2015). This algorithm estimates DMSPt as a function of chlorophyll *a* concentration (Chl) by switching between two different equations depending on the light penetration regime, defined by the quotient between euphotic layer depth (Zeu) and mixed layer depth (MLD),

$\log_{10}$DMSPt = 1.70 + 1.14 $\log_{10}$Chl + 0.44 $\log_{10}$Chl$^2$ + 0.063 SST - 0.0024 SST$^2$,

Zeu/MLD $\geq$ 1, *stratified* water column      (eq. A1)

$\log_{10}$DMSPt = 1.74 + 0.81 $\log_{10}$Chl + 0.60 $\log_{10}$(Zeu/MLD),

10     Zeu/MLD < 1, *mixed* water column      (eq. A2)

with Chl in mg m$^{-3}$ (= µg L$^{-1}$) and sea surface temperature (SST) in °C. This scheme implicitly reproduces the seasonal ecological succession from low-DMSP phytoplankton taxa (mainly diatoms) towards high- DMSP taxa (mainly haptophytes and dinoflagellates) that are relatively more abundant in stratified conditions, when the entire sea-surface mixed layer is well

15     illuminated. To avoid uncertain algorithm behavior at extreme Chl concentrations, Chl$_{SAT}$ lower (higher) than 0.04 (60) mg m$^{-3}$ was set to these respective values. The DMSPt$_{SAT}$ sub-algorithm uses a third equation, based on satellite-retrieved PIC, in coccolith laden waters where Chl algorithms fail (Galí et al., 2015). In practice, this equation is not used in climatological implementations (see Fig. 2).

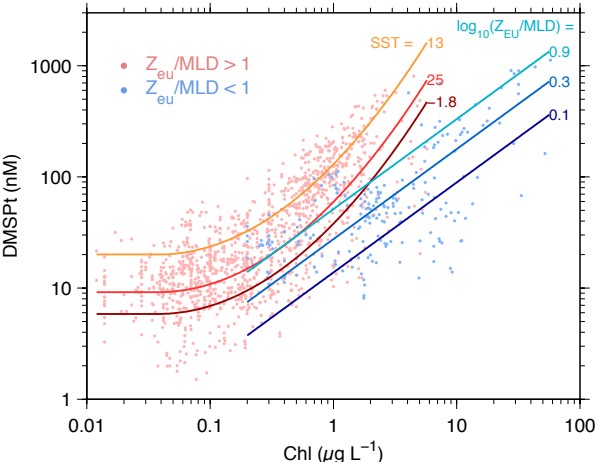

**Figure A1: DMSPt$_{SAT}$ sub-algorithm.** Dots in the background show database measurements, and lines represent fits for "stratified" and "mixed" conditions at different SST and Z$_{eu}$/MLD ratios, respectively.