# Peer review of "Diagnosing sea-surface dimethylsulfide (DMS) concentration from satellite data at global and regional scales"

_Biogeosciences, 2018_

## Referee Comment (RC1) · Anonymous Referee #1 · 13 Mar 2018

This manuscript, entitled "Diagnosing sea-surface dimethylsulfide (DMS) concentration from satellite data at global and regional scales", used climatological MLD, satellite re-trieved Chl and PAR, and an embedded sub-algorithm based on satellite data to con-struct an algorithm for surface DMS concentrations. This work provides a prescribed DMS distribution with seasonal and interannual variability. The paper is well written and well organized. Yet I recommend the following points to be clarified or modified.

Main comments:

1. One major contribution of this algorithm is providing DMS estimate with interannual and seasonal variability. Authors chose some regions to show the variability, but the

results are part of the validation and not representative as mentioned in the manuscript (being the region where the algorithm works the best). It would be better if authors can discuss more about the variability on a global scale.

2. Authors discussed regional tuning and biases as the strength of the algorithm. However, it raises the question about predictive power. In other words, the algorithm is largely built based on statistical regression, lacking fundamental scientific support. Authors should further clarify the optimized formula, differences caused by regional tuning, and regional tuning is required in some cases.

3. The algorithm discussed here largely depend on the sub-algorithm. Though it is described in a previous publication, basic introduction and discussion about the sub-algorithm are needed for readers to understand the strength and limitation. For example, chlorophyll data contains no information about speciation, which plays an important role in the total DMS concentration.

Specific comments: P2, L31: Please add reference

P3, L5: Since diatom-dominated blooms produce low DMSP per unit biomass, how to determine the equation parameters based on chlorophyll concentrations alone? It's clear that blooms dominated by different phytoplankton will require different parameter values, though the chlorophyll levels are similar.

P4. L30: please explain more the motivation of using PAR, instead of the total short-wave irradiance. Using PAR here imply the role of photosynthesis, however it shouldn't play any role in the DMSP to DMS transformation. Though mentioned a bit in the comparison with VS07, it's not clear whether and why the choice in the present work is better.

P 5. L 24: Conclusions about the comparison are vague. More discussion about various choices would be helpful.

P8: Please discuss uncertainties attributable to DMSP estimates. P8 sec 3.1.2 Authors

showed how DMS estimates vary with parameter values, but not physical meanings, which should be included.

P9 L26: estimated DMS concentrations are much lower than L11. Does it suggest an overestimation in L11? Please clarify.

P10 L26 Please explain the cause of the disagreement.

P12: Different regions require different sets of parameter values for optimization, which raises the question about uncertainties. Please elaborate on it.

P13: This section discusses biases of the algorithm without explanation of causes and suggestions on potential improvements.

---

## Short Comment (SC1) · 15 Mar 2018

Response to MAIN COMMENTS only

Reviewer: 1. One major contribution of this algorithm is providing DMS estimate with interannual and seasonal variability. Authors chose some regions to show the variability, but the results are part of the validation and not representative as mentioned in the manuscript (being the region where the algorithm works the best). It would be better if authors can discuss more about the variability on a global scale.

Author: I thank the reviewer for appreciating our contribution. As he/she indicates, a

global-scale analysis of interannual variability would be an interesting exercise, and one that we plan to do soon. However, we declined this possibility in the present paper for the following reasons: (i) we thought of this paper as a proof-of-concept, and it is already quite long; (ii) running the algorithm for >10 years of satellite observations with the appropriate temporal resolution is doable but not trivial in terms of data storage and processing capacity; (iii) as described in the paper, we already implemented the algorithm for the MODIS-Aqua 2003-2016 record for latitudes >45N (at daily 4.6 km resolution), but the major results of this analysis will be analyzed elsewhere; (iv) an analysis of interannual variability of the seasonal cycle is most informative in coherent ecoregions where sufficient real DMS data are available for validation. In my opinion, there are two possible rigorous approaches to that: analyzing variability in regions/stations where in situ time series exist (which we did: BATS and OSP stations in Fig. 10); and analyzing variability in regions where the algorithm shows very good skill, evaluated both in a "scatterplot view" and in a "seasonal view", lending more credit to the satellite-diagnosed patterns (which we also did: Fig. 9 and Fig. S3).

I would also like to stress that the algorithm works well in regions other than the temperate and subpolar North Atlantic. I attached two figures (Fig. R1.1 and R1.2) showing that the algorithm works even better in the Bering Sea, which is well documented regarding in situ DMS data (see map in https://saga.pmel.noaa.gov/dms/). The figures correspond to areas of size similar to those shown in Fig. 9 of the paper and have the same legend. Both were derived from MODIS-Aqua data.

Reviewer: 2. Authors discussed regional tuning and biases as the strength of the algorithm. However, it raises the question about predictive power. In other words, the algorithm is largely built based on statistical regression, lacking fundamental scientific support. Authors should further clarify the optimized formula, differences caused by regional tuning, and regional tuning is required in some cases.

Author: We completely disagree with R#1 about the algorithm "lacking fundamental scientific support", ad we will make this clearer in the revised version of the paper. The

effects of sunlight on DMS production-consumption budgets have been experimentally demonstrated by several studies (Archer et al., 2010; Galí et al., 2013a, 2013b, 2013c; Royer et al., 2016; Toole et al., 2006). Although UVB and UVA elicit the strongest responses, PAR can also stimulate plankton DMS production(Archer et al., 2010; Galí et al., 2013c). More importantly, since incident PAR and UVR are strongly correlated on a global scale, satellite-retrieved PAR is an excellent first-order approximation for UVR effects.

The reasons why solar PAR+UVR irradiance drive the DMS seasonal cycle were extensively discussed by Galí and Simó (2015). Basically, at high irradiance, there is (i) a higher proportion of high-DMSP phytoplankton species, (ii) a higher community DMSP-to-DMS conversion yield, (iii) an increase in DMS photolysis rate constants, and (iv) a decrease in bacterial consumption rate constants. Factors (i) and (ii) synergistically combine to increase gross DMS production rates, whereas factors (iii) and (iv) compensate each other so that total DMS removal rate constants do not change as much as gross DMS production. As a result, DMS budgets imply a higher "equilibrium" DMS concentration during high irradiance seasons (DMS is at quasi steady state on daily-to-weekly time scales most of the time, see also Royer et al. 2016). These are the robust theoretical underpinnings of our algorithm.

As a corollary, community DMSPt-to-DMS yields are significantly correlated to the DMS/DMSPt ratio. We attached a further figure to illustrate this (Fig. R1.3). The figure is based on the same global-ocean DMS(P) cycling process database analyzed by Galí and Simó (2015).

This also (partially) responds the specific comment "P4".

Reviewer: 3. The algorithm discussed here largely depends on the sub-algorithm. Though it is described in a previous publication, basic introduction and discussion about the sub- algorithm are needed for readers to understand the strength and limitation. For example, chlorophyll data contains no information about speciation, which

plays an important role in the total DMS concentration.

Author: We will include a brief description of the DMSPt sub-algorithm in the revised version of the paper, perhaps in annex (we assume R#1 refers to the DMSPt sub-algorithm). Although this algorithm, thoroughly described and validate by Galí et al. 2015, does not include explicit phytoplankton speciation, it implicitly discriminates different types of phytoplankton communities.

Briefly, the DMSPt sub-algorithm is based on 2 equations that predict DMSPt from Chl and other secondary variables. The algorithm switches between these 2 equations depending on a classical bio-optical criterion, the ratio Zeu/MLD (Zeu is euphotic layer depth defined by 1% surface PAR penetration; MLD is mixed layer depth) (Uitz et al., 2006). * Zeu/MLD >1 indicates "stratified waters" where the mixed layer is entirely well illuminated. In these conditions, phytoplankton communities have higher proportions of DMSP-rich taxa, mainly haptophytes but also dinoflagellates, and other picoeukaryotes with generally lower abundance (chrystophytes, pelagopphytes, prasinophytes). * Zeu/MLD < 1 indicates more deeply "mixed waters" where part of the mixed layer is below the 1% irradiance level. In these conditions, DMSP-poor phytoplankton (mostly diatoms) dominates.

In accordance, at a given Chl concentration, the "stratified-waters " DMSPt equation produces a tenfold higher sea-surface DMSPt concentration (approximately) than the "mixed-waters" equation. Detailed information on the DMSPt sub-algorithm can be found in Galí et al. (2015), which is freely and legally available (after the 2 year embargo) on ResearchGate: https://www.researchgate.net/profile/Marti_Gali_Tapias/contributions.

This also responds the specific comment "P3".

References Archer, S. D., Ragni, M., Webster, R., Airs, R. L. and Geider, R. J.: Dimethyl sulfoniopropionate and dimethyl sulfide production in response to photoinhibition in Emiliania huxleyi, Limnol. Oceanogr., 55(4), 1579–1589,

[Figure]

doi:10.4319/lo.2010.55.4.1579, 2010. Galí, M. and Simó, R.: A meta-analysis of oceanic DMS and DMSP cycling processes: Disentangling the summer paradox, Global Biogeochem. Cycles, 29, 496–515, doi:10.1002/2014GB004940, 2015. Galí, M., Simó, R., Vila‐Costa, M., Ruiz‐González, C., Gasol, J. M. and Matrai, P. A.: Diel patterns of oceanic dimethylsulfide (DMS) cycling: Microbial and physical drivers, Global Biogeochem. Cycles, 27, 620–636, 2013a. Galí, M., Simó, R., Pérez, G. L., Ruiz-González, C., Sarmento, H., Fuentes-Lema, A., Royer, S.-J. and Gasol, J. M.: Differential response of planktonic primary, bacterial, and dimethylsulfide production rates to vertically-moving and static incubations in upper mixed-layer summer sea waters, Biogeosciences, 10, 7983–7998, 2013b. Galí, M., Ruiz-González, C., Lefort, T., Gasol, J. M., Cardelús, C., Romera-Castillo, C. and Simó, R.: Spectral irradiance dependence of sunlight effects on plankton dimethylsulfide production, Limnol. Oceanogr., 58(2), 489–504, 2013c. Galí, M., Devred, E., Levasseur, M., Royer, S.-J. and Babin, M.: A remote sensing algorithm for planktonic dimethylsulfoniopropionate (DMSP) and an analysis of global patterns, Remote Sens. Environ., 171, 171–184, doi:10.1016/j.rse.2015.10.012, 2015. Royer, S.-J., Galí, M., Mahajan, A. S., Ross, O. N., Pérez, G. L., Saltzman, E. S. and Simó, R.: A high-resolution time-depth view of dimethylsulphide cycling in the surface sea, Sci. Rep., (August), 1–13, doi:10.1038/srep32325, 2016. Toole, D. A., Slezak, D., Kiene, R. P., Kieber, D. J. and Siegel, D. A.: Effects of solar radiation on dimethylsulfide cycling in the western Atlantic Ocean, Deep Sea Res. Part I, 53, 136–153, 2006. Uitz, J., Claustre, H., Morel, A. and Hooker, S. B.: Vertical distribution of phytoplankton communities in open ocean: An assessment based on surface chlorophyll, J. Geophys. Res., 111(C8), doi:10.1029/2005JC003207, 2006.

[Figure]

**Fig. 1.** DMS mean seasonal cycle and interannual variability in the Bering Sea (central shelf). Latitude 59-62N, longitude 169-174W (see text for details). Colors: years; black: mean; gray: L11 climatology

[Figure]

**Fig. 2.** As Fig. 1 for the coastal Bering Sea shelf. Latitude 54-57N, longitude 164-168W (see text for details). Colors: years; black: mean; gray: L11 climatology

[Figure]

**Fig. 3.** Relationship between process-based DMS production yield from total DMSP and the DMS/DMSPt concentration ratio (see text for further details)

---

## Referee Comment (RC2) · Anonymous Referee #2 · 19 Mar 2018

General Comments:

The manuscript "Diagnosing sea-surface dimethylsulfide (DMS) concentration from satellite data at global and regional scales" highlights the need to improve spatial and temporal scale of DMS through satellite data sets. The science community has relied on the available climatology data for couple of years. However, validity of current data sets (Kettle et al., 1999; Lana et al., 2011) are limited because of significant uncertainty and lack of interannual variability. I think the algorithm presented in this manuscript coupled with preceding studies on DMSSAT are very important to assess the challenges with global and regional DMS data sets. The algorithm proposed relies on the nonlinear relationship between phytoplankton light exposure, DMSP and DMS. Again this research is a step in the right direction to produce appropriate observation-related data of reduced sulfur from the ocean that could be employed to constrain Earth System Models (ESMs). Suitable DMS dataset could ultimately help to reduce uncertainty associated with the impact of tropospheric aerosol forcing on global radiation. However, satellite generated dataset are not immune from uncertainties. Thus, there is a general concern of compounding uncertainty transferred to derived dataset such as $DMSP_{SAT}$ and $DMS_{SAT}$. Algorithm and dataset presented in this manuscript could therefore be improved in different biomes. The paper addresses relevant scientific question and the overall presentation is well structured and clear. I encourage the paper to be published after addressing the following concerns:

Main Comments: 1) Global $DMS_{SAT}$ concentration seems a little bit low. In-situ measurements at BATS station in late summer to early spring of 2006 and 2007 show higher concentration than $DMS_{SAT}$ from equation 2f. With this regional underestimation, it will be reasonable to assume that area weighted global mean DMS obtained from equation 2f (Table 3) could also be a little bit underestimated. Thus, An annual emission of 16–18 Tg S yr$^{-1}$ could significantly lower the formation of sulfate aerosol in ESMs below atmospheric measurements at the boundary layer. Could you combine optimize local scale concentrations (for BATS and any available region) with the global scale to improve the overall concentrations?

3) Authors should discuss if any extrapolation method was used to compute DMS concentration at high latitudes where SeaWIFS chlorophyll concentrations are limited. If none, then authors should be careful to note the spatial coverage of DMS in the winter. Otherwise, authors could also be quantitative on the overall polar concentrations reported in L11 climatology but missing in this study due to limitations in satellite chlorophyll measurements.

5) The authors should clarify why they computed global $DMS_{SAT}$ fields with SeaWIFS data and regional DMSSAT with MODIS data. I was wondering if the authors made

a global scale optimization (MLongh) with MODIS data? Monthly DMS climatologies derive from $DMS_{SAT}$ (MODIS-Aqua) tend to be more agreeable with in-situ measurements. The authors should report how the global climatology computed from MODIS differs from SeaWIFS and/or Lana et al. 2011. Specific Comments:

Page 2, line 32 needs reference "...10% is emitted to the atmosphere through turbulent diffusion [ref]".

Page 3, line 6 needs reference "... and low DMS yield [ref]".

Page 4, line 15: I have the impression you used MODIS-Aqua (2003-2016) for the satellite matchups. Please clarify if you used 2003-2012 data for something else.

Page 5, line 17: which environmental variables?

Write out what N means in table 1

Figure 3b: In the caption, do you mean DMS/DMSPt ratio vs PAR (or vs DMSPt)?

Figure 7: Caption for $DMS_{SAT}$ algorithm should be (b)

---

## Author Comment (AC1) · 13 Apr 2018

We posted our response to R1 and R2 as a supplement that includes (i) a general response and overview of proposed changes, (ii) detailed responses to R1, and (iii) detailed responses to R2.

Please also note the supplement to this comment:
https://www.biogeosciences-discuss.net/bg-2018-18/bg-2018-18-AC1-supplement.pdf

[Figure]

[Figure]

Bering

**Fig. 1.**

[Figure]

**Fig. 2.**

[Figure]

**Fig. 3.**

**Supplement:**

**General response and overview of proposed changes**

**Authors**: We thank the two referees for appreciating our contribution and for their encouraging comments. Their observations prompted us to conduct a more detailed analysis of bias in our algorithm, which we will include in the new manuscript. This analysis deepened our understanding of the behavior of the algorithm, and further convinced us of the validity of our approach. In response to the referee's concerns we would like to propose the changes summarized below:

1. Extending the abstract to make it more comprehensive and informative, following Biogeosciences recommendations. We propose including the general form of eq. 2 (which links DMS to DMSPt and PAR) while highlighting the strong mechanistic basis of our empirical formulation (response to R1, general comment #2). If possible, we would like to cite in the abstract our previous work describing the DMSPt sub-algorithm.

2. Including, for completeness, an annex with a brief description of the DMSPt sub-algorithm, which was described in depth by Galí et al. (2015) (response to R1 general comment #3).

3. Reshaping the third and the last paragraph of the Introduction to clarify that our two-step empirical algorithm has a sound mechanistic basis (response to R1 general comment #2).

4. Further clarifying and justifying the primary satellite datasets used to produce each of the $DMS_{SAT}$ datasets (Methods subsections 2.1 and 2.3) and the corresponding in situ data used to validate them (Results subsection 3.1). These changes address criticisms from R2 (response to general comments #2 and #3) and minor comments from R1 (regarding the use of PAR and the propagation of uncertainties from the DMSPt sub-algorithm).

5. Better organizing subsection 3.1.2 to clarify the physical meaning of eq. 2 parameters (in response to R1 comment).

6. Further justifying our choices regarding algorithm configurations for the global, the regional and the local scales. Since both reviewers are concerned with potential negative bias in our global $DMS_{SAT}$ climatology, we propose adding a new table with a detailed bias assessment (response to R2 general comment #1), and discussing its implications (Discussion). The new table shows that, excluding the Southern Ocean and the Coastal biomes (where $Chl_{SAT}$ causes a negative and positive $DMS_{SAT}$ bias, respectively), the mean bias of $DMS_{SAT}$ in the remainder of the global ocean is likely -16% to -20%, at

most. This suggests that the interpolation based DMS climatology of Lana et al. (2011) is biased in a similar proportion (around 15%), adding to the statistical evidence already given in section 4.1 and Fig. 8. We also note that this finding may have implications for our understanding of DMS effects on CCN. Since DMS control of marine boundary layer CCN populations is well established (Quinn et al., 2017), the efficiency with which DMS promotes gas-to-particle conversion might need upward revision.

7. Correcting errors in figure 3 (R2 spotted an error in the caption), but also figure 2 (text in red and blue boxes was exchanged) and figure 4 (we spotted a minor error in the data subsets used for the calculation of statistics; it does not affect the validity of the results and alters only slightly the algorithm skill statistics).

On the other hand, we would like to decline the following recommendations:

1. Assessing interannual variability in DMS concentrations at the global scale (as proposed by R1). The main objective of our paper is presenting a new approach to estimate sea-surface DMS, and the example datasets are sufficient for a proof-of-concept, in our judgment (response to R1 general comment #1). An analysis of DMS concentration and emission variability at latitudes >45N will be presented elsewhere (Galí et al., in prep.).

2. Producing a new global climatology with regionally variable model coefficients. While acknowledging the regional biases in our algorithm, we defend the interest and validity of our global-scale estimates. Factoring regional variability into the global scale algorithm to resolve "endogenous error" is not a trivial problem, and correcting for the $Chl_{SAT}$ bias to resolve "exogenous error" is not the matter of our paper (response to R2 general comment #1 and R1 specific comment).

Detailed reasons for our choices are given in responses below.

**Response to Anonymous Referee #1** (comment received and published 13 March 2018)

**R1**: This manuscript, entitled "Diagnosing sea-surface dimethylsulfide (DMS) concentration from satellite data at global and regional scales", used climatological MLD, satellite retrieved Chl and PAR, and an embedded sub-algorithm based on satellite data to construct an algorithm for surface DMS concentrations. This work provides a prescribed DMS distribution with seasonal and interannual variability. The paper is well written and well organized. Yet I recommend the following points to be clarified or modified.

Main comments:
1. One major contribution of this algorithm is providing DMS estimate with interannual and seasonal variability. Authors chose some regions to show the variability, but the results are part of the validation and not representative as mentioned in the manuscript (being the region where the algorithm works the best). It would be better if authors can discuss more about the variability on a global scale.

Authors: We agree with R1 that a global-scale analysis of interannual variability would be an interesting exercise, and one that we plan to do in the future. However, we declined this possibility in the present paper for the following reasons: (i) we thought of this paper as a proof-of-concept, and it is already quite long (~8500 words with planned modifications); (ii) running the algorithm for >10 years of satellite observations with the appropriate temporal resolution (at least 8 days) is doable but not trivial in terms of data storage and processing capacity; (iii) as described in the paper, we already implemented the algorithm for the MODIS-Aqua 2003-2016 record for latitudes >45N (at daily 4.6 km resolution), and the major results of this analysis will be analyzed elsewhere (manuscript in preparation); (iv) an analysis of interannual variability of the seasonal cycle is most informative in coherent ecoregions where sufficient in situ DMS data are available for validation. In our opinion, there are two possible rigorous approaches to address this: analyzing variability in regions/stations where in situ time series exist (which we did: BATS and OSP stations in Fig. 10); and analyzing variability in regions where the algorithm shows very good skill, evaluated both in a "scatterplot view" and in a "seasonal view", lending more credit to the satellite-diagnosed patterns (which we also did: Fig. 9 and Fig. S3).

Besides, we would like to stress that the algorithm optimized for the region >45N works well in regions other than the temperate and subpolar North Atlantic. We attached two figures (Fig. R1 and R2) showing that the algorithm works very well in the Bering Sea, a region where in situ DMS concentrations are well documented (see map in https://saga.pmel.noaa.gov/dms/). The figures correspond to areas of size similar to those shown in Fig. 9 of the paper and have the same legend.

Reviewer:
2. Authors discussed regional tuning and biases as the strength of the algorithm. However, it raises the question about predictive power. In other words, the algorithm is largely built based on statistical regression, lacking fundamental scientific support. Authors should further clarify the optimized formula, differences caused by regional tuning, and regional tuning is required in some cases.

Authors: We disagree with R1 about the algorithm "lacking fundamental scientific support". The effects of sunlight on DMS production-consumption budgets have been experimentally demonstrated by several studies (Archer et al., 2010; Galí et al., 2013a, 2013b, 2013c; Royer et al., 2016; Toole et al., 2006). Although UVB and UVA elicit the strongest responses, PAR can also stimulate plankton DMS production (Archer et al., 2010; Galí et al., 2013c). More importantly, since clouds are the main atmospheric attenuators in the visible and UV regions, incident PAR and UVR are strongly correlated (Bordewijk et al., 1995; Calbó et al., 2005), and satellite-retrieved PAR is an excellent first-order approximation for UVR effects. Moreover, seawater transparency in the UVR is also strongly correlated to that in the PAR region in most oceanic waters, where phytoplankton-derived materials drive sunlight attenuation in the water column, further strengthening the coherence between PAR and UVR.

The reasons why solar PAR+UVR irradiance drive the DMS seasonal cycle were extensively discussed by Galí and Simó (2015). Basically, at high irradiance, there is (i) a higher proportion of high-DMSP phytoplankton species, (ii) a higher community DMSP-to-DMS conversion yield, (iii) an increase in DMS photolysis rate constants, and (iv) a decrease in bacterial consumption rate constants. Factors (i) and (ii) synergistically combine to increase gross DMS production rates, whereas factors (iii) and (iv) compensate each other so that total DMS removal rate constants do not change as much as gross DMS production. As a result, DMS budgets imply a higher near-steady-state DMS concentration during high irradiance seasons. (Note that DMS is at near-steady-state on daily-to-weekly time scales most of the time, see Galí and Simó (2015) and Royer et al. (2016)). These are the robust theoretical underpinnings of our algorithm. Statistical fitting of in situ data is used to translate this mechanistic knowledge into model parameters with predictive value.

As a corollary, community DMSPt-to-DMS yields are significantly correlated to the DMS/DMSPt ratio. We attached a further figure to illustrate this (Fig. R3). The figure is based on the same global-ocean DMS(P) cycling process database analyzed by Galí and Simó (2015), and could be added as Annex.

In response to R1 criticism, we will briefly review the information provided above in the Introduction of the revised manuscript. Regarding the clarity of the formulas used: the paper is already clear regarding the formula used to produce each dataset. Formulas are indicated throughout the text (eq. 2a-h), compiled in Table 2, and indicated in Fig. 10 where more than one formula was used.

Reviewer:

3. The algorithm discussed here largely depends on the sub-algorithm. Though it is described in a previous publication, basic introduction and discussion about the sub-algorithm are needed for readers to understand the strength and limitation. For example, chlorophyll data contains no information about speciation, which plays an important role in the total DMS concentration.

Authors: We propose adding an annex with a brief description of the DMSPt sub-algorithm in the revised version of the paper (we assume R#1 refers to the DMSPt sub-algorithm). Although this algorithm, thoroughly described and validated by Galí et al. 2015, does not include explicit phytoplankton speciation, it implicitly discriminates different types of phytoplankton communities.

Briefly, the DMSPt sub-algorithm is based on two equations that predict DMSPt from Chl and other secondary variables. The algorithm switches between these equations depending on a classical bio-optical criterion, the ratio Zeu/MLD (Zeu is euphotic layer depth defined by 1% surface PAR penetration; MLD is mixed layer depth) (Uitz et al., 2006).
* Zeu/MLD >1 indicates "stratified waters" where the mixed layer is entirely well illuminated. In these conditions, phytoplankton communities have higher proportions of DMSP-rich taxa, mainly haptophytes but also dinoflagellates, and other picoeukaryotes with generally lower abundance (chrystophytes, pelagopphytes, prasinophytes).
* Zeu/MLD < 1 indicates more deeply "mixed waters" where part of the mixed layer is below the 1% irradiance level. In these conditions, DMSP-poor phytoplankton (mostly diatoms) dominates.

At a given Chl concentration, the "stratified-waters " DMSPt equation produces a tenfold higher sea-surface DMSPt concentration (approximately) than the "mixed-waters" equation. Detailed information on the DMSPt sub-algorithm can be found in Galí et al. (2015), which is freely and legally available (after the 2 year embargo) on ResearchGate: https://www.researchgate.net/profile/Marti_Gali_Tapias/contributions.

Specific comments:

P2, L31: Please add reference
A: We will cite Galí & Simó (2015).

P3, L5: Since diatom-dominated blooms produce low DMSP per unit biomass, how to determine the equation parameters based on chlorophyll concentrations alone? It's clear that blooms dominated by different phytoplankton will require different parameter values, though the chlorophyll levels are similar.

A: Please see response to general comment 3.

P4. L30: please explain more the motivation of using PAR, instead of the total short-wave irradiance. Using PAR here imply the role of photosynthesis, however it shouldn't play any role in the DMSP to DMS transformation. Though mentioned a bit in the comparison with VS07, it's not clear whether and why the choice in the present work is better.
A: Please see response to general comment 2. We will add a short explanation at the end of subsection 2.1 in the new version.

P 5. L 24: Conclusions about the comparison are vague. More discussion about various choices would be helpful.
A: Lines 23-24 in page 5, which belong to section 2 (Methods), currently read:
"Based on the correlation analysis, we built several regression models where DMS was predicted as a function of in situ DMSPt concentration and one or more additional variables (Table 1)".

We are not sure we understand the point made by R2. The variables used as additional predictors in the stepwise regression are those listed in Table 1, and the results of the stepwise regression are briefly described in section 3.1.1 (Results, Statistical exploration). A compilation of regression models and statistics is given in Table S4, and the potential predictive power of additional variables is discussed in section 4.2 (Discussion, 4.2 How far can we go with empirical remote sensing algorithms?).

Although we devoted much effort to find regression models that improved on our base model (eq. 2), none of the several tested models provided robust and significant improvements. This result is the main conclusion of section 3.1.1, and is the reason why we chose to be concise when it came to describing our (comprehensive) statistical explorations. Readers are referred to Table S4, which can be a useful starting point for future studies that may benefit from a larger DMS database.

At the editor's request, we can revise the paragraph quoted above (P5, L23-30), the description of the stepwise regression results (section 3.1.1) and their discussion in section 4.2.

P8: Please discuss uncertainties attributable to DMSP estimates.

This comment is addressed in section 3.1.3, page 9. We decline in-depth analyses of the uncertainty associated with the DMSPt sub-algorithm alone for the sake of concision and for the following reasons: (i) the DMSPt sub-algorithm was already thoroughly validated

by Galí et al. (2015); (ii) we find it more useful to assess uncertainty in the complete, two-step algorithm, which partly results from uncertainty in satellite Chl that propagates through the DMSPt sub-algorithm; (iii) the amount of in situ DMSPt measurements available is smaller than that of in situ Chl (page 4 line 10), so that validation controlled for $Chl_{SAT}$ error is statistically more powerful; and (iv) in situ DMSPt measurements were already used to fit the algorithm equations, so they have to be excluded to achieve and independent validation. All these ideas were already expressed in page 9 lines 9-15, and in consequence we decline modifying the text in this regard.

P8 sec 3.1.2 Authors showed how DMS estimates vary with parameter values, but not physical meanings, which should be included.

Following R2 recommendation, we propose reorganizing section 3.1.2 (although the physical meaning of model coefficients was already described in the former manuscript). The physical meaning of the model eq. 2 coefficients will be clearly described in a separate, along with their interval of variation according to bootstrapped regression. The revised section would be organized following this sequence of ideas:

1. Introductory sentence (as it is). Introduce Table 2 and bootstrap analysis here (formerly described in page 8 lines 25-26.
2. Alpha coefficient: intercept.
3. Beta coefficient: The DMSPt coefficient can be seen as a biomass-dependent modulation of DMSP-to-DMS conversion efficiency, probably reflecting planktonic food web structure and biogeography.
4. Gamma coefficient: The PAR coefficient can be seen as a DMSPt-independent sensitivity of DMSPt-to-DMS conversion efficiency.
5. Examples of typical DMS/DMSPt ratios across ocean regimes and DMSPt and PAR levels as formerly described in page 8 lines 15-22.
6. Interrelationships among eq. 2 coefficients based on bootstrap analysis, as formerly described in page 8 lines 26-31.

At the editor's request, we can add additional panels in figure 3 to show how changing eq. 2 coefficients impacts DMS retrievals.

P9 L26: estimated DMS concentrations are much lower than L11. Does it suggest an overestimation in L11? Please clarify.
A: Discussion of algorithm bias with respect to L11 does not belong to this section. The benchmark for validation is real data, not L11. As mentioned in the general response and, which much more detail, in the response to R2 general comment #1, the disagreement between L11 and $DMS_{SAT}$ has multiple causes, and most of them were already discussed

in sections 4.1 and 4.2. Section 4.1 focused on well-known causes of positive bias in L11 (sparse sampling, interpolation/extrapolation procedures), whereas 4.2 focused on built-in ("endogenous") limitations of our approach (negative bias caused by inability to produce high DMS/DMSPt ratios under certain conditions and in specific regions).

In summary, neither L11 nor DMSSAT are without limitations. The exact value of mean DMS concentration and emission fields is not known, and it is logical that different statistical reconstruction approaches lead to different estimates. When evaluated using the full DMS dataset for the satellite era (later than 1997), $DMS_{SAT}$ has a negative bias of -9%. This bias compounds a bias of -16 to -20% in most of the ocean, a likely Chl-driven positive bias in coastal areas, and a Chl-driven negative bias in the southern polar waters. We will add a new table with a detailed description of the bias assessment in the revised version of the manuscript, and the related information will beadded to the Discussion.

P10 L26 Please explain the cause of the disagreement.
A: The disagreement is probably explained by a strong negative bias in satellite Chl as already reported at the end of section 4.1 (and mentioned in the abstract, as well).

P12: Different regions require different sets of parameter values for optimization, which raises the question about uncertainties. Please elaborate on it.
A: The need for regional tuning is commonplace in the development of ocean color remote sensing algorithms, and in particular at high latitudes (Cota et al., 2004; Johnson et al., 2013; Morel and Gentili, 2009; Ben Mustapha et al., 2012); but the well-acknowledged need for regional algorithms does not prevent space agencies from implementation of global-scale optimized algorithms for products such as Chl, Kd, etc. Regional tuning must rely on both a solid theoretical understanding of regional particularities and data availability.

As argued above, our algorithm relies on robust understanding of the links between light exposure, plankton biogeography and DMS budgets. These relationships do not suffice to accurately predict DMS in some regions/seasons, as we discuss in subsection 4.2, but neither the exact causes nor the geographic extent of regional anomalies are well known. As shown in the new Table (see response to R2), data are too scarce in the southern polar ocean to allow for development of robust regional algorithms. Conversely, we were able to develop a robust regional algorithm for high northern latitudes (>45N) thanks to much better coverage in the database.

As indicated in page 16 lines 20-21 (section 4.2), " Tuning the eq. 2 coefficients is a workable alternative to better reproduce the mean seasonal cycle in certain regions (Fig. 10), and eq. 2 could perhaps be generalized in a way that allowed its coefficients to vary

across different biogeochemical regimes". However, doing so is not trivial if one wants to avoid discontinuities at the border of biogeographic regions. Besides data availability and theoretical understanding, there is another strong reason for a single global scale optimized model: global data provides the largest possible dynamic ranges in all variables, helping avoid overfitting.

P13: This section discusses biases of the algorithm without explanation of causes and suggestions on potential improvements.
A: Explanations of the causes of algorithm biases (and other shortcomings) and suggestions on potential improvements were given in section 4.2 (see response to previous question). We concluded that significant improvements are unlikely using a small amount of predictor variables (as we did). Future improvements will probably rely on much better spatial-temporal coverage of in the global DMS database and new process studies, leading to better understanding of regional DMS drivers.

**Response to Anonymous Referee #2**

General Comments:
The manuscript "Diagnosing sea-surface dimethylsulfide (DMS) concentration from satellite data at global and regional scales" highlights the need to improve spatial and temporal scale of DMS through satellite data sets. The science community has relied on the available climatology data for couple of years. However, validity of current data sets (Kettle et al., 1999; Lana et al., 2011) are limited because of significant uncertainty and lack of interannual variability. I think the algorithm presented in this manuscript coupled with preceding studies on DMSSAT are very important to assess the challenges with global and regional DMS data sets. The algorithm proposed relies on the non-linear relationship between phytoplankton light exposure, DMSP and DMS. Again this research is a step in the right direction to produce appropriate observation-related data of reduced sulfur from the ocean that could be employed to constrain Earth System Models (ESMs). Suitable DMS dataset could ultimately help to reduce uncertainty associated with the impact of tropospheric aerosol forcing on global radiation. However, satellite generated dataset are not immune from uncertainties. Thus, there is a general concern of compounding uncertainty transferred to derived dataset such as DMSPSAT and DMSSAT. Algorithm and dataset presented in this manuscript could therefore be improved in different biomes. The paper addresses relevant scientific question and the overall presentation is well structured and clear. I encourage the paper to be published after addressing the following concerns:

Main Comments:

1) Global DMSSAT concentration seems a little bit low. In-situ measurements at BATS station in late summer to early spring of 2006 and 2007 show higher concentration than DMSSAT from equation 2f. With this regional underestimation, it will be reasonable to assume that area weighted global mean DMS obtained from equation 2f (Table 3) could also be a little bit underestimated. Thus, An annual emission of 16–18 Tg S yr−1 could significantly lower the formation of sulfate aerosol in ESMs below atmospheric measurements at the boundary layer. Could you combine optimize local scale concentrations (for BATS and any available region) with the global scale to improve the overall concentrations?

Authors: We agree with R2 that our global DMS may be underestimated. However, the global underestimation is not necessarily driven by the tropical and subtropical oceans, as one would think after a quick look at the BATS time series (Fig. 10). We prepared a table to show in more details the bias of $DMS_{SAT}$ across ocean biomes when compared to in situ DMS data or to the L11 climatology. In situ database comparisons were made using all available matchups and also by constraining the error in $Chl_{SAT}$ with respect to in situ Chl. Validation using satellite matchups with unconstrained $Chl_{SAT}$ error indicates $DMS_{SAT}$ has a small positive bias (+6%) in the Trades biome, and a -14% bias when $Chl_{SAT}$ error is constrained. Thus, the -34% difference between $DMS_{SAT}$ and L11 in that biome probably results from both $DMS_{SAT}$ underestimation and L11 overestimation. More generally, this exercise shows that the assessment of algorithm bias is not unequivocal, as it depends on the combination of known sources of bias ($Chl_{SAT}$) and unknown sources of bias (e.g., how representative are of an entire biome a few tens or hundreds of available measurements?).

**New Table X**. Relative deviation ("bias") of the global-scale optimized $DMS_{SAT}$ algorithm across ocean biomes when evaluated against the database, using satellite matchups with (a) constrained $Chl_{SAT}$ error with respect to $Chl_{IN\ SITU}$, or (b) all matchups, and (c) against the L11 monthly climatology. The three criteria are used for a qualitative overall assessment of $DMS_{SAT}$ bias (last column), classifying the most likely magnitude and sign as small± (±10%), moderate (10 to 40%, either + or -) or large (>40%, either + or -).

| Biome | Database matchups | | | | | c) L11 gridded data | | DMS$_{SAT}$ bias assessment |
|---|---|---|---|---|---|---|---|---|
| | a) Chl$_{SAT}$ RMSE < 0.3 | | | b) All matchups | | | | |
| | Bias DMS$_{SAT}$ | Bias Chl$_{SAT}$ | N[a] | Bias DMS$_{SAT}$ | N[a] | Bias DMS$_{SAT}$ | Area[b] | |
| Polar N | -28% | -7% | 54 | -39% | 2291 | -35% | 3.0% | Moderate- |
| Westerlies N | -4% | -11% | 125 | -13% | 1872 | -34% | 10.1% | Small± |
| Trades | -14% | 7% | 442 | 6% | 3489 | -34% | 56.6% | Small± to moderate- |
| Westerlies S[c] | -3% | -30% | 30 | -27% | 1585 | -26% | 14.2% | Small± to moderate- |
| Polar S | -59% | -6% | 42 | -47% | 170 | -76% | 5.7% | Large- |
| Coastal | 46% | 4% | 377 | 3% | 5673 | -6% | 10.4% | Small± to large+? |
| Global ocean | 11% | 2% | 1053 | -9% | 15080 | -36% | 100% | Small± to moderate- |
| Global ocean -pS -C | -16% | -2% | 634 | -20% | 9249 | -33% | 83.9% | Moderate- |

[a]For the database comparisons, the amount of data available for validation in a given region depends on the total amount of measurements, the proportion of data points with satellite matchups and, for the constrained case, the fraction of matchups where Chl$_{SAT}$ has log$_{10}$ RMSE < 0.5 compared to concurrent Chl$_{IN\ SITU}$. Samples with available DMSPt measurements were excluded because they were used in model fitting and optimization.
[b]For the global climatology comparison we report the % of ocean area, excluding pixels that could not be observed by satellites (high latitude winter).
[c]Samples where DMSPt was measured not excluded for this biome due to the small amount of data. Removing them would leave N = 13.

Further analysis of this table shows that most of the underestimation in the global scale optimized algorithm occurs in northern and southern high latitudes. In the paper we show that, in high northern latitudes, this gap can be filled by developing a specific parameterization, which benefits from the relatively abundant in situ data and the use of GSM algorithm to correct for continental interferences in the retrieval of Chl$_{SAT}$. A similar exercise would be extremely interesting in the Southern Ocean, but the small number of DMS and DMSPt samples and concurrent satellite matchups precludes robust estimation of the algorithm coefficients.

The high latitude underestimation seems to be compensated by overestimation in coastal waters, yielding a global scale bias of -9% (all matchups) to +11% (constrained Chl$_{SAT}$ error). As already reported in the ms (e.g. Table 3), using semi-analytical ocean color algorithms should produce more robust results in optically complex, continentally affected waters. Note also that these statistics are subject to the disproportionate influence of coastal measurements.

If we exclude southern polar and coastal waters, constrained and unconstrained assessments of the DMS$_{SAT}$ converge at -16 to -20%. This assessment, combined with the

compelling case for a high-DMS bias in the database (section 4.1, Fig. 8), allows us to conclude that the L11 climatology provides an upper bound (likely an overestimate) of global DMS concentrations, and the $DMS_{SAT}$ a lower bound (likely an underestimate). In consequence, attempts to improve $DMS_{SAT}$ should not strive to match L11.

Assuming that $DMS_{SAT}$ has a global mean bias of -10%, we could easily correct it by increasing $\alpha$ in eq 2f by $\log_{10}(1.1)$, e.g. from -1.237 to -1.196. However, this would not improve the representation of the mean seasonal cycle in areas like BATS or OSP.
As indicated in page 16 lines 20-21 (section 4.2), " Tuning the eq. 2 coefficients is a workable alternative to better reproduce the mean seasonal cycle in certain regions (Fig. 10), and eq. 2 could perhaps be generalized in a way that allowed its coefficients to vary across different biogeochemical regimes". However, doing so is not trivial if one wants to avoid discontinuities at the border of biogeographic regions. Since we conceived this paper as a proof-of-concept, we found sufficient to produce a global-scale optimized dataset, a regionally optimized dataset (>45N, including OSP) and a locally optimized dataset (BATS) for illustrative purposes. By discussing the limitations of each dataset, we gained insight into factors, other than light exposure, that may control the mean DMS seasonal cycle and its interannual variability in different regions.

Finally, regarding R2's argument that lower global mean DMS "could significantly lower the formation of sulfate aerosol in ESMs below atmospheric measurements at the boundary layer": this argument can easily be reversed. Given that independent atmospheric observations support the dominant role of DMS in controlling marine aerosol and CCN populations (Quinn et al., 2017), lower marine concentration and emission would mean that atmospheric chemistry modules in ESM's underestimate the efficiency with which DMS nucleates new aerosol particles that grow to CCN relevant sizes.

2) Authors should discuss if any extrapolation method was used to compute DMS concentration at high latitudes where SeaWIFS chlorophyll concentrations are limited. If none, then authors should be careful to note the spatial coverage of DMS in the winter. Otherwise, authors could also be quantitative on the overall polar concentrations reported in L11 climatology but missing in this study due to limitations in satellite chlorophyll measurements.

Authors: We did not use any extrapolation method to compute DMS outside the satellite observed areas. Results shown in Fig. 5 are the means of December through February (Northern hemisphere) or June through August (Southern hemisphere) and unobserved pixels were left blank. The northward (southward) latitude limit of ocean color satellite observation is at about 48 degrees in December (June), 52 degrees in January (July) and

60 degrees in February (August). Therefore, most observations during the winter season at latitude 52-60 degrees represent the month of February (August) only.

DMS concentrations at high latitudes attain their lowest annual values in winter, which probably corresponds to their weakest effects on aerosols and clouds. Therefore, there is little interest in obtaining full DMS data coverage in winter at high latitudes. We will include this information in section 2.3 (Algorithm implementation).

Finally, we would like to note that our satellite diagnosed DMS concentrations at high northern latitudes in the winter season (DJF) are at odds with those estimated through interpolation by Lana et al. (2011), and probably more realistic than their estimates. Analysis of available DMS data shows that this in an interpolation artifact in the L11 climatology. L11 suggests a latitudinal increase in mean zonal DMS in winter, from about 0.6 nM at 50N to >1.5 nM at 80N (Figure 5a). However, the DJF mean ± std of database DMS between 45N and 60N is 0.63 ± 0.42 (median of 0.53) N = 1136, and no measurements are available north of 60N for the DJF months. In addition, existing data represent mostly continental shelves (median depth of 160 m). A similar case can be made for March and April.

3) The authors should clarify why they computed global $DMS_{SAT}$ fields with SeaWIFS data and regional DMSSAT with MODIS data. I was wondering if the authors made a global scale optimization (MLongh) with MODIS data? Monthly DMS climatologies derive from $DMS_{SAT}$ (MODIS-Aqua) tend to be more agreeable with in-situ measurements. The authors should report how the global climatology computed from MODIS differs from SeaWIFS and/or Lana et al. 2011.

Authors: We computed global $DMS_{SAT}$ with SeaWiFS because the SeaWiFS climatology, based on the period 1998-2010, overlaps in time with 55% of the in situ database measurements used to develop the L11 climatology, whereas the MODIS-Aqua record (2003-present) temporal overlap reaches only 41% of those measurements. Thus, using SeaWiFS maximizes the temporal overlap to compare the $DMS_{SAT}$ and L11 climatologies. This being said, DMS climatologies derived from SeaWiFS and MODIS-Aqua would be very similar. We already compared the global monthly DMSPt climatologies derived from these two sensors in our previous work (Galí et al., 2015), where we reported (section 5.1, bottom of page 177 of that paper) that "the difference between global DMSPt climatologies derived from SeaWiFS and MODIS-Aqua is generally negligible: 74% of pixels differ by less than ±5% and the global mean difference is 1 ± 9%".

Indeed, our algorithm coefficients can be tuned to maximize the model-data fit depending on the satellite sensor used. However, since our previous work showed that differences

among satellite sensors are a relatively small source of uncertainty, we found more useful to optimize a single set of coefficients for both sensors. The justification of merging SeaWiFS and MODIS matchup, which already appeared in SI section S2, can be moved to the main text.

Finally, R2 seems to assume that MODIS-derived $DMS_{SAT}$ is better than the one derived from SeaWiFS, although we did not report any direct comparison between SeaWiFS- and MODIS-derived $DMS_{SAT}$. The direct comparison can be done for those measurements where nearly simultaneous SeaWiFS and MODIS matchups exist. But again, this is a minor source of uncertainty.

Specific Comments:

Page 2, line 32 needs reference "...10% is emitted to the atmosphere through turbulent diffusion [ref]".

A: We will cite Galí & Simó (2015).

Page 3, line 6 needs reference "... and low DMS yield [ref]".

A: We will cite Lizotte et al. (2012).

Page 4, line 15: I have the impression you used MODIS-Aqua (2003-2016) for the satellite matchups. Please clarify if you used 2003-2012 data for something else.

A: As clearly explained in that paragraph, we merged SeaWiFS and MODIS matchups

Page 5, line 17: which environmental variables? Write out what N means in table 1

A: Will be corrected. Environmental variables were listed in Table 1. N is sample size.

Figure 3b: In the caption, do you mean DMS/DMSPt ratio vs PAR (or vs DMSPt)?

A: Will be corrected, we meant vs. DMSPt.

Figure 7: Caption for $DMS_{SAT}$ algorithm should be (b)

A: Will be corrected.

**References**

Archer, S. D., Ragni, M., Webster, R., Airs, R. L. and Geider, R. J.: Dimethyl sulfoniopropionate and dimethyl sulfide production in response to photoinhibition in Emiliania huxleyi, Limnol. Oceanogr., 55(4), 1579–1589, doi:10.4319/lo.2010.55.4.1579, 2010.

Bordewijk, J. A., Slaper, H., Reinen, H. A. J. M. and Schlamann, E.: Total solar radiation and the influence of clouds and aerosols on the biologically effective UV, Geophys. Res. Lett., 22(16), 2151–2154, 1995.

Calbó, J., Pagès, D. and González, J.-A.: Empirical studies of cloud effects on UV radiation: A review, Rev. Geophys., 43(RG2002), 1–28, doi:10.1029/2004RG000155.1.INTRODUCTION, 2005.

Cota, G. F., Wang, J. and Comiso, J. C.: Transformation of global satellite chlorophyll retrievals with a regionally tuned algorithm, Remote Sens. Environ., 90, 373–377, doi:10.1016/j.rse.2004.01.005, 2004.

Galí, M. and Simó, R.: A meta-analysis of oceanic DMS and DMSP cycling processes: Disentangling the summer paradox, Global Biogeochem. Cycles, 29, 496–515, doi:10.1002/2014GB004940, 2015.

Galí, M., Simó, R., Vila‑Costa, M., Ruiz‑González, C., Gasol, J. M. and Matrai, P. A.: Diel patterns of oceanic dimethylsulfide (DMS) cycling: Microbial and physical drivers, Global Biogeochem. Cycles, 27, 620–636, 2013a.

Galí, M., Simó, R., Pérez, G. L., Ruiz-González, C., Sarmento, H., Fuentes-Lema, A., Royer, S.-J. and Gasol, J. M.: Differential response of planktonic primary, bacterial, and dimethylsulfide production rates to vertically-moving and static incubations in upper mixed-layer summer sea waters, Biogeosciences, 10, 7983–7998, 2013b.

Galí, M., Ruiz-González, C., Lefort, T., Gasol, J. M., Cardelús, C., Romera-Castillo, C. and Simó, R.: Spectral irradiance dependence of sunlight effects on plankton dimethylsulfide production, Limnol. Oceanogr., 58(2), 489–504, 2013c.

Galí, M., Devred, E., Levasseur, M., Royer, S.-J. and Babin, M.: A remote sensing algorithm for planktonic dimethylsulfoniopropionate (DMSP) and an analysis of global patterns, Remote Sens. Environ., 171, 171–184, doi:10.1016/j.rse.2015.10.012, 2015.

Johnson, R., Strutton, P. G., Wright, S. W., McMinn, A. and Meiners, K. M.: Three improved satellite chlorophyll algorithms for the Southern Ocean, J. Geophys. Res. Ocean., 118(7), 3694–3703, doi:10.1002/jgrc.20270, 2013.

Morel, A. and Gentili, B.: A simple band ratio technique to quantify the colored dissolved and detrital organic material from ocean color remotely sensed data, Remote Sens. Environ., 113(5), 998–1011, doi:10.1016/j.rse.2009.01.008, 2009.

Ben Mustapha, S., Bélanger, S. and Larouche, P.: Evaluation of ocean color algorithms in the southeastern Beaufort Sea, Canadian Arctic: New parameterization using SeaWiFS, MODIS, and MERIS spectral bands, Can. J. Remote Sens., 8992(July 2014), 535–556, doi:10.5589/m12-045, 2012.

Quinn, P. K., Coffman, D. J., Johnson, J. E., Upchurch, L. M. and Bates, T. S.: Small fraction of marine cloud condensation nuclei made up of sea spray aerosol, Nat. Geosci., 10(August), doi:10.1038/ngeo3003, 2017.

Royer, S.-J., Galí, M., Mahajan, A. S., Ross, O. N., Pérez, G. L., Saltzman, E. S. and Simó, R.: A high-resolution time-depth view of dimethylsulphide cycling in the

surface sea, Sci. Rep., (August), 1–13, doi:10.1038/srep32325, 2016.

Toole, D. A., Slezak, D., Kiene, R. P., Kieber, D. J. and Siegel, D. A.: Effects of solar radiation on dimethylsulfide cycling in the western Atlantic Ocean, Deep Sea Res. Part I, 53, 136–153, 2006.

Uitz, J., Claustre, H., Morel, A. and Hooker, S. B.: Vertical distribution of phytoplankton communities in open ocean: An assessment based on surface chlorophyll, J. Geophys. Res., 111(C8), doi:10.1029/2005JC003207, 2006.

---

## Author Response (AR1)

**General response and overview of the changes**

**Authors**: We thank the two referees for appreciating our contribution and for their encouraging comments. Their observations prompted us to conduct a more detailed analysis of bias in our algorithm, which we included in the new manuscript. This analysis deepened our understanding of the behavior of the algorithm, and further convinced us of the validity of our approach. In addition to changes prompted by reviewers' criticisms, we also improved some figures. All the changes are listed below:

1. We extended the abstract to make it more comprehensive and informative, following Biogeosciences recommendations. We included the general form of eq. 2 (which links DMS to DMSPt and PAR) while highlighting the strong mechanistic basis of our empirical formulation (response to R1, general comment #2). We cite in the abstract our previous work describing the DMSPt sub-algorithm (Galí et al., 2015).

2. We added Appendix A, with a brief description of the DMSPt sub-algorithm, which was described in depth by Galí et al. (2015) (response to R1 general comment #3).

3. We edited paragraphs 5 and 7 of the Introduction to clarify that our two-step empirical algorithm has a sound mechanistic basis (response to R1 general comment #2). We made some smaller edits in the Introduction.

4. We further clarified and justified the primary satellite datasets used to produce each of the DMS$_{SAT}$ datasets (Methods subsections 2.1 and 2.3, Results subsection 3.1.3). These changes addressed criticisms from R2 (response to general comments #2 and #3) and minor comments from R1 (regarding the use of PAR and the propagation of uncertainties from the DMSPt sub-algorithm). We tried to shorten the Methods section and improve the flow wherever possible (e.g., subsection 2.2).

5. We improved subsection 3.1.2 to clarify the physical meaning of eq. 2 parameters (in response to R1 comment).

6. We further justified our choices regarding algorithm configurations for the global, the regional and the local scales. Since both reviewers were concerned with negative bias in our global DMS$_{SAT}$ climatology, we added a new table (now Table 3) with a detailed bias assessment (response to R2 general comment #1). We also briefly described this bias assessment in subsection 3.1.3 and discussed its causes and consequences in the Discussion and the Conclusions. The new table shows that, excluding the Southern Ocean and the Coastal biomes (where Chl$_{SAT}$ causes a negative and positive DMS$_{SAT}$ bias, respectively), the mean bias of DMS$_{SAT}$ in the remainder of the global ocean is likely -16% to -20%, at most. This suggests that the interpolation based DMS climatology of Lana et al. (2011) is biased in a similar proportion (around 15%), adding to the statistical evidence already given in section 4.1 and Fig. 8.

7. We corrected errors in Fig. 3 (caption), Fig. 2 (text in red and blue boxes was exchanged) and Fig. 4 (minor error in the data subsets used for the calculation of statistics; it does not affect the validity of the results and alters only slightly the algorithm skill statistics).

The following changes were not prompted by the reviewers:

8. We added three introductory paragraphs in the Discussion, before subsection 4.1, reshaping some pieces of text that used to appear later in the same section. While the argument flow of the Discussion remains the same, we tried to improve the

writing and strengthen the lines of evidence supporting our conclusions. We also made minor edits in the Conclusions to refine our message.

9. Changes in figures
- Fig. 1: we added a panel with monthly in situ data availability in 5x5 degree bins.
- Fig. 6 (old Fig. 7): we added Hovmoller diagrams for mean and median in situ DMS, and a boxplot with the bin mean/median ratios.
- Fig. 7 (old Fig. 8): we added histograms comparing in situ DMS to L11, $DMS_{SAT}$, SD02 and VS07 climatologies.

On the other hand, we declined the following recommendations:

1. Assessing interannual variability in DMS concentrations at the global scale (as proposed by R1). The main objective of our paper is presenting a new approach to estimate sea-surface DMS, and the example datasets are sufficient for a proof-of-concept, in our judgment (response to R1 general comment #1). An analysis of DMS concentration and emission variability at latitudes >45N will be presented elsewhere (Galí et al., in prep.).

2. Producing a new global climatology with regionally variable model coefficients. While acknowledging the regional biases in our algorithm, we defend the interest and validity of our global-scale estimates. Factoring regional variability into the global scale algorithm to resolve "endogenous error" is not a trivial problem, and correcting for the $Chl_{SAT}$ bias to resolve "exogenous error" is not the matter of our paper (response to R2 general comment #1 and R1 specific comment).

Detailed point-by point responses to the reviewers were given in the followin document:
https://editor.copernicus.org/index.php/bg-2018-18-AC1.pdf?_mdl=msover_md&_jrl=11&_lcm=oc108lcm109w&_acm=get_comm_file&_ms=66128&c=140392&salt=675281 3041594696979

[revised manuscript text omitted]

Martí Galí Tàpias 2018-5-8 9:36

Martí Galí Tàpias 2018-5-3 14:30

Martí Galí Tàpias 2018-5-7 22:50

Martí Galí Tàpias 2018-5-2 12:37
Deleted: . As illustrated in Fig. 6, mean DMS concentration in a given month and biome is systematically higher than the corresponding median. In most biomes, $DMS_{SAT}$ tends to follow the monthly medians of in situ data, whereas $DMS_{L11}$ generally follows –by construction– the monthly means.

Martí Galí Tàpias 2018-5-7 17:01
Moved down [1]: Since $DMS_{SAT}$ has a small positive or negative bias when validated on non-binned data (Table S5), our analysis suggests that the L11 climatology and its predecessors suffer a global positive bias.

Martí Galí Tàpias 2018-5-2 16:13

Martí Galí Tàpias 2018-5-3 14:44

[revised manuscript text omitted]

(Lavoie et al., 2015; Spiese et al., 2015)

| Page 5: [1] Deleted | Martí Galí Tàpias | 2018-04-25 15:25 |
|---|---|---|

(Lavoie et al., 2015; Spiese et al., 2015)

| Page 5: [2] Deleted | Martí Galí Tàpias | 2018-04-25 15:44 |
|---|---|---|

This

| Page 5: [2] Deleted | Martí Galí Tàpias | 2018-04-25 15:44 |
|---|---|---|

This

| Page 5: [2] Deleted | Martí Galí Tàpias | 2018-04-25 15:44 |
|---|---|---|

This

| Page 5: [2] Deleted | Martí Galí Tàpias | 2018-04-25 15:44 |
|---|---|---|

This

| Page 5: [2] Deleted | Martí Galí Tàpias | 2018-04-25 15:44 |
|---|---|---|

This

| Page 5: [2] Deleted | Martí Galí Tàpias | 2018-04-25 15:44 |
|---|---|---|

This

| Page 5: [2] Deleted | Martí Galí Tàpias | 2018-04-25 15:44 |
|---|---|---|

This

| Page 5: [2] Deleted | Martí Galí Tàpias | 2018-04-25 15:44 |
|---|---|---|

This

| Page 5: [2] Deleted | Martí Galí Tàpias | 2018-04-25 15:44 |
|---|---|---|

This

| Page 5: [2] Deleted | Martí Galí Tàpias | 2018-04-25 15:44 |
|---|---|---|

This

| Page 5: [2] Deleted | Martí Galí Tàpias | 2018-04-25 15:44 |
|---|---|---|

This

| Page 5: [2] Deleted | Martí Galí Tàpias | 2018-04-25 15:44 |
|---|---|---|

This

| Page 5: [3] Deleted | Martí Galí Tàpias | 2018-05-03 14:59 |
|---|---|---|

therefore

| Page 5: [3] Deleted | Martí Galí Tàpias | 2018-05-03 14:59 |
|---|---|---|

therefore

| Page 5: [3] Deleted | Martí Galí Tàpias | 2018-05-03 14:59 |
|---|---|---|

therefore

| Page 5: [3] Deleted | Martí Galí Tàpias | 2018-05-03 14:59 |
|---|---|---|

therefore

| Page 5: [3] Deleted | Martí Galí Tàpias | 2018-05-03 14:59 |
|---|---|---|

therefore

| Page 5: [3] Deleted | Martí Galí Tàpias | 2018-05-03 14:59 |
|---|---|---|

therefore

| Page 5: [3] Deleted | Martí Galí Tàpias | 2018-05-03 14:59 |
|---|---|---|

therefore

| Page 5: [3] Deleted | Martí Galí Tàpias | 2018-05-03 14:59 |
|---|---|---|

therefore

| Page 5: [3] Deleted | Martí Galí Tàpias | 2018-05-03 14:59 |
|---|---|---|

therefore

| Page 5: [3] Deleted | Martí Galí Tàpias | 2018-05-03 14:59 |
|---|---|---|

therefore

| Page 5: [3] Deleted | Martí Galí Tàpias | 2018-05-03 14:59 |
|---|---|---|

therefore

| Page 6: [4] Deleted | Martí Galí Tàpias | 2018-04-26 10:56 |
|---|---|---|

exhibit large discrepancies

| Page 6: [4] Deleted | Martí Galí Tàpias | 2018-04-26 10:56 |
|---|---|---|

exhibit large discrepancies

| Page 6: [4] Deleted | Martí Galí Tàpias | 2018-04-26 10:56 |
|---|---|---|

exhibit large discrepancies

| Page 6: [4] Deleted | Martí Galí Tàpias | 2018-04-26 10:56 |
|---|---|---|

exhibit large discrepancies

| Page 6: [4] Deleted | Martí Galí Tàpias | 2018-04-26 10:56 |
|---|---|---|

exhibit large discrepancies

| Page 6: [4] Deleted | Martí Galí Tàpias | 2018-04-26 10:56 |
|---|---|---|

exhibit large discrepancies

| Page 6: [5] Deleted | Martí Galí Tàpias | 2018-04-25 16:27 |
| --- | --- | --- |

empirical

| Page 6: [6] Deleted | Martí Galí Tàpias | 2018-05-15 18:34 |
| --- | --- | --- |

satellite matchup data and

| Page 6: [7] Deleted | Martí Galí Tàpias | 2018-05-15 18:33 |
| --- | --- | --- |

, equivalent to $\mu g\ L^{-1}$

| Page 6: [7] Deleted | Martí Galí Tàpias | 2018-05-15 18:33 |
|---|---|---|

, equivalent to μg L$^{-1}$

| Page 10: [8] Deleted | Martí Galí Tàpias | 2018-05-14 12:09 |
|---|---|---|

one new

| Page 10: [9] Deleted | Martí Galí Tàpias | 2018-05-15 19:08 |
|---|---|---|

the predictive power

| Page 10: [9] Deleted | Martí Galí Tàpias | 2018-05-15 19:08 |

the predictive power

| Page 11: [10] Formatted | Martí Galí Tàpias | 2018-05-14 12:22 |

Subscript

| Page 11: [11] Deleted | Martí Galí Tàpias | 2018-05-07 10:23 |

First, it must be noted that the $\log_{10}$DMSPt coefficient (
.

| Page 11: [12] Formatted | Martí Galí Tàpias | 2018-05-14 12:22 |

Font:Symbol

| Page 11: [13] Deleted | Martí Galí Tàpias | 2018-05-07 11:43 |

this

| Page 11: [13] Deleted | Martí Galí Tàpias | 2018-05-07 11:43 |
|---|---|---|

this

| Page 11: [13] Deleted | Martí Galí Tàpias | 2018-05-07 11:43 |
|---|---|---|

this

| Page 11: [13] Deleted | Martí Galí Tàpias | 2018-05-07 11:43 |
|---|---|---|

this

| Page 11: [13] Deleted | Martí Galí Tàpias | 2018-05-07 11:43 |
|---|---|---|

this

| Page 11: [14] Deleted | Martí Galí Tàpias | 2018-05-14 12:22 |
|---|---|---|

Second, we note that the y-intercept ($\alpha$), the $\log_{10}$DMSPt coefficient ($\beta$) and the PAR coefficient ($\gamma$) vary in a consistent manner as the binning spatial scale increases (Table 2).

| Page 11: [14] Deleted | Martí Galí Tàpias | 2018-05-14 12:22 |
|---|---|---|

Second, we note that the y-intercept ($\alpha$), the $\log_{10}$DMSPt coefficient ($\beta$) and the PAR coefficient ($\gamma$) vary in a consistent manner as the binning spatial scale increases (Table 2).

| Page 11: [15] Deleted | Martí Galí Tàpias | 2018-05-15 19:09 |
|---|---|---|

predictive

| Page 11: [15] Deleted | Martí Galí Tàpias | 2018-05-15 19:09 |
|---|---|---|

predictive

| Page 11: [15] Deleted | Martí Galí Tàpias | 2018-05-15 19:09 |
|---|---|---|

predictive

| Page 12: [16] Deleted | Martí Galí Tàpias | 2018-05-07 11:59 |
|---|---|---|

-

| Page 12: [16] Deleted | Martí Galí Tàpias | 2018-05-07 11:59 |
|---|---|---|

-

| Page 12: [16] Deleted | Martí Galí Tàpias | 2018-05-07 11:59 |
|---|---|---|

-

| Page 12: [16] Deleted | Martí Galí Tàpias | 2018-05-07 11:59 |
|---|---|---|

-

| Page 12: [17] Deleted | Martí Galí Tàpias | 2018-05-07 12:01 |
|---|---|---|

| Page 12: [17] Deleted | Martí Galí Tàpias | 2018-05-07 12:01 |
|---|---|---|

| | | |
|---|---|---|
| **Page 12: [17] Deleted** | **Martí Galí Tàpias** | **2018-05-07 12:01** |
| **Page 12: [17] Deleted** | **Martí Galí Tàpias** | **2018-05-07 12:01** |
| **Page 12: [17] Deleted** | **Martí Galí Tàpias** | **2018-05-07 12:01** |
| **Page 12: [18] Deleted** | **Martí Galí Tàpias** | **2018-05-04 12:41** |
| **Page 12: [18] Deleted** | **Martí Galí Tàpias** | **2018-05-04 12:41** |
| **Page 12: [18] Deleted** | **Martí Galí Tàpias** | **2018-05-04 12:41** |
| **Page 12: [18] Deleted** | **Martí Galí Tàpias** | **2018-05-04 12:41** |
| **Page 12: [18] Deleted** | **Martí Galí Tàpias** | **2018-05-04 12:41** |
| **Page 12: [18] Deleted** | **Martí Galí Tàpias** | **2018-05-04 12:41** |
| **Page 12: [18] Deleted** | **Martí Galí Tàpias** | **2018-05-04 12:41** |
| **Page 12: [18] Deleted** | **Martí Galí Tàpias** | **2018-05-04 12:41** |
| **Page 12: [18] Deleted** | **Martí Galí Tàpias** | **2018-05-04 12:41** |
| **Page 12: [18] Deleted** | **Martí Galí Tàpias** | **2018-05-04 12:41** |
| **Page 12: [18] Deleted** | **Martí Galí Tàpias** | **2018-05-04 12:41** |
| **Page 12: [18] Deleted** | **Martí Galí Tàpias** | **2018-05-04 12:41** |
| **Page 12: [18] Deleted** | **Martí Galí Tàpias** | **2018-05-04 12:41** |

| Page 12: [18] Deleted | Martí Galí Tàpias | 2018-05-04 12:41 |
|---|---|---|

| Page 12: [18] Deleted | Martí Galí Tàpias | 2018-05-04 12:41 |
|---|---|---|

| Page 12: [18] Deleted | Martí Galí Tàpias | 2018-05-04 12:41 |
|---|---|---|

| Page 12: [18] Deleted | Martí Galí Tàpias | 2018-05-04 12:41 |
|---|---|---|

| Page 12: [18] Deleted | Martí Galí Tàpias | 2018-05-04 12:41 |
|---|---|---|

| Page 12: [18] Deleted | Martí Galí Tàpias | 2018-05-04 12:41 |
|---|---|---|

| Page 12: [18] Deleted | Martí Galí Tàpias | 2018-05-04 12:41 |
|---|---|---|

| Page 12: [18] Deleted | Martí Galí Tàpias | 2018-05-04 12:41 |
|---|---|---|

| Page 12: [18] Deleted | Martí Galí Tàpias | 2018-05-04 12:41 |
|---|---|---|

| Page 12: [18] Deleted | Martí Galí Tàpias | 2018-05-04 12:41 |
|---|---|---|

| Page 13: [19] Deleted | Martí Galí Tàpias | 2018-05-11 17:51 |
|---|---|---|

After verifying the good performance of the algorithm,

| Page 13: [19] Deleted | Martí Galí Tàpias | 2018-05-11 17:51 |
|---|---|---|

After verifying the good performance of the algorithm,

| Page 13: [19] Deleted | Martí Galí Tàpias | 2018-05-11 17:51 |
|---|---|---|

After verifying the good performance of the algorithm,

| Page 13: [19] Deleted | Martí Galí Tàpias | 2018-05-11 17:51 |
|---|---|---|

After verifying the good performance of the algorithm,

| Page 13: [19] Deleted | Martí Galí Tàpias | 2018-05-11 17:51 |
|---|---|---|

After verifying the good performance of the algorithm,

| Page 13: [19] Deleted | Martí Galí Tàpias | 2018-05-11 17:51 |
|---|---|---|

After verifying the good performance of the algorithm,

| Page 13: [19] Deleted | Martí Galí Tàpias | 2018-05-11 17:51 |
|---|---|---|

After verifying the good performance of the algorithm,

| Page 13: [19] Deleted | Martí Galí Tàpias | 2018-05-11 17:51 |

After verifying the good performance of the algorithm,

| Page 13: [19] Deleted | Martí Galí Tàpias | 2018-05-11 17:51 |

After verifying the good performance of the algorithm,

| Page 13: [19] Deleted | Martí Galí Tàpias | 2018-05-11 17:51 |

After verifying the good performance of the algorithm,

| Page 13: [19] Deleted | Martí Galí Tàpias | 2018-05-11 17:51 |

After verifying the good performance of the algorithm,

| Page 13: [19] Deleted | Martí Galí Tàpias | 2018-05-11 17:51 |

After verifying the good performance of the algorithm,

| Page 13: [20] Deleted | Martí Galí Tàpias | 2018-05-07 14:47 |

, as well as

| Page 13: [20] Deleted | Martí Galí Tàpias | 2018-05-07 14:47 |

, as well as

| Page 13: [21] Deleted | Martí Galí Tàpias | 2018-05-01 11:39 |

| Page 13: [21] Deleted | Martí Galí Tàpias | 2018-05-01 11:39 |

| Page 13: [22] Deleted | Martí Galí Tàpias | 2018-05-14 14:23 |

As shown in Fig. 5-7,

| Page 13: [22] Deleted | Martí Galí Tàpias | 2018-05-14 14:23 |

As shown in Fig. 5-7,

| Page 13: [22] Deleted | Martí Galí Tàpias | 2018-05-14 14:23 |

As shown in Fig. 5-7,

| Page 13: [22] Deleted | Martí Galí Tàpias | 2018-05-14 14:23 |

As shown in Fig. 5-7,

| Page 13: [22] Deleted | Martí Galí Tàpias | 2018-05-14 14:23 |

As shown in Fig. 5-7,

| Page 13: [22] Deleted | Martí Galí Tàpias | 2018-05-14 14:23 |

As shown in Fig. 5-7,

| Page 13: [22] Deleted | Martí Galí Tàpias | 2018-05-14 14:23 |

As shown in Fig. 5-7,

| Page 13: [22] Deleted | Martí Galí Tàpias | 2018-05-14 14:23 |

As shown in Fig. 5-7,

| Page 15: [23] Deleted | Martí Galí Tàpias | 2018-05-14 14:28 |

| Page 15: [24] Deleted | Martí Galí Tàpias | 2018-05-02 10:58 |

We selected the subpolar North Atlantic because it is one of the regions where the algorithm works best (Fig. S3), lending credit to observed variability patterns.

| Page 15: [24] Deleted | Martí Galí Tàpias | 2018-05-02 10:58 |
|---|---|---|

We selected the subpolar North Atlantic because it is one of the regions where the algorithm works best (Fig. S3), lending credit to observed variability patterns.

| Page 15: [25] Deleted | Martí Galí Tàpias | 2018-05-02 11:08 |
|---|---|---|

The

| Page 15: [25] Deleted | Martí Galí Tàpias | 2018-05-02 11:08 |
|---|---|---|

The

| Page 15: [25] Deleted | Martí Galí Tàpias | 2018-05-02 11:08 |
|---|---|---|

The

| Page 16: [26] Deleted | Martí Galí Tàpias | 2018-05-02 11:10 |
|---|---|---|

ed

| Page 16: [26] Deleted | Martí Galí Tàpias | 2018-05-02 11:10 |
|---|---|---|

ed

| Page 16: [26] Deleted | Martí Galí Tàpias | 2018-05-02 11:10 |
|---|---|---|

ed

| Page 16: [27] Deleted | Martí Galí Tàpias | 2018-05-02 11:46 |
|---|---|---|

Note

| Page 16: [27] Deleted | Martí Galí Tàpias | 2018-05-02 11:46 |
|---|---|---|

Note

| Page 16: [27] Deleted | Martí Galí Tàpias | 2018-05-02 11:46 |
|---|---|---|

Note

| Page 16: [27] Deleted | Martí Galí Tàpias | 2018-05-02 11:46 |
|---|---|---|

Note

| Page 16: [27] Deleted | Martí Galí Tàpias | 2018-05-02 11:46 |
|---|---|---|

Note

| Page 16: [27] Deleted | Martí Galí Tàpias | 2018-05-02 11:46 |
|---|---|---|

Note

| Page 16: [28] Deleted | Martí Galí Tàpias | 2018-05-04 15:18 |
|---|---|---|

d

| Page 16: [28] Deleted | Martí Galí Tàpias | 2018-05-04 15:18 |
|---|---|---|

d

| Page 16: [28] Deleted | Martí Galí Tàpias | 2018-05-04 15:18 |
|---|---|---|

d

| Page 16: [28] Deleted | Martí Galí Tàpias | 2018-05-04 15:18 |
|---|---|---|

d

| Page 16: [28] Deleted | Martí Galí Tàpias | 2018-05-04 15:18 |
|---|---|---|

d

| Page 16: [28] Deleted | Martí Galí Tàpias | 2018-05-04 15:18 |
|---|---|---|

d

| Page 16: [28] Deleted | Martí Galí Tàpias | 2018-05-04 15:18 |
|---|---|---|

d

| Page 16: [28] Deleted | Martí Galí Tàpias | 2018-05-04 15:18 |
|---|---|---|

d

| Page 16: [28] Deleted | Martí Galí Tàpias | 2018-05-04 15:18 |
|---|---|---|

d

| Page 16: [28] Deleted | Martí Galí Tàpias | 2018-05-04 15:18 |
|---|---|---|

d

| Page 16: [28] Deleted | Martí Galí Tàpias | 2018-05-04 15:18 |
|---|---|---|

d

| Page 16: [28] Deleted | Martí Galí Tàpias | 2018-05-04 15:18 |
|---|---|---|

d

| Page 16: [28] Deleted | Martí Galí Tàpias | 2018-05-04 15:18 |
|---|---|---|

d

| Page 16: [29] Deleted | Martí Galí Tàpias | 2018-05-07 15:15 |
|---|---|---|

 (Fig. 8D)

| Page 16: [29] Deleted | Martí Galí Tàpias | 2018-05-07 15:15 |
|---|---|---|

 (Fig. 8D)

| Page 16: [29] Deleted | Martí Galí Tàpias | 2018-05-07 15:15 |
|---|---|---|

 (Fig. 8D)

| Page 16: [29] Deleted | Martí Galí Tàpias | 2018-05-07 15:15 |
|---|---|---|

 (Fig. 8D)

| Page 16: [29] Deleted | Martí Galí Tàpias | 2018-05-07 15:15 |
|---|---|---|

 (Fig. 8D)

| Page 17: [30] Deleted | Martí Galí Tàpias | 2018-05-07 16:34 |
|---|---|---|

(

| Page 17: [30] Deleted | Martí Galí Tàpias | 2018-05-07 16:34 |
|---|---|---|

(

| Page 17: [31] Deleted | Martí Galí Tàpias | 2018-05-07 16:02 |

Here

| Page 17: [32] Deleted | Martí Galí Tàpias | 2018-05-02 12:38 |

**Geo-statistics, remote sensing algorithms and**

| Page 17: [33] Deleted | Martí Galí Tàpias | 2018-05-07 17:07 |

In our view, the reasons for the disagreement are many fold

| Page 17: [34] Deleted | Martí Galí Tàpias | 2018-05-03 14:29 |

First, t

First, t

| **Page 18: [35] Deleted** | **Martí Galí Tàpias** | **2018-05-08 9:36** |

for

| **Page 18: [36] Deleted** | **Martí Galí Tàpias** | **2018-05-03 14:30** |

Second, s

| **Page 18: [37] Deleted** | **Martí Galí Tàpias** | **2018-05-07 22:50** |

(

| Page 18: [37] Deleted | Martí Galí Tàpias | 2018-05-07 22:50 |
|---|---|---|

(

| Page 18: [38] Deleted | Martí Galí Tàpias | 2018-05-03 14:44 |
|---|---|---|

The third major issue is t

| Page 18: [38] Deleted | Martí Galí Tàpias | 2018-05-03 14:44 |

The third major issue is t

| Page 19: [39] Deleted | Martí Galí Tàpias | 2018-05-07 17:01 |

| Page 19: [40] Deleted | Martí Galí Tàpias | 2018-05-14 15:04 |

.

| Page 19: [41] Formatted | Martí Galí Tàpias | 2018-05-03 15:52 |

Highlight

| Page 19: [42] Deleted | Martí Galí Tàpias | 2018-05-07 17:18 |

The $DMS_{SAT}$ algorithm captures in situ variability (Fig. 4) using a small set of predictor variables (Fig. 2). Moreover, it reproduces the mismatch between DMS and Chl such that, at a given $Chl_{SAT}$ concentration, diagnosed DMS can vary by up to 40-fold (Fig. 8D). This mismatch is stronger than that produced by the SD02 or the VS07 algorithms. The correlation between $DMS_{SAT}$ and $Chl_{SAT}$ is 0.34 in the global climatology, similar to that found in the global database (r = 0.39), and perhaps more realistic than that between the $DMS_{L11}$ climatology and the SeaWiFS Chl climatology (r = 0.15) (Fig. 8). Another positive feature of our algorithm is its capacity to produce a DMS peak in summer across different latitudes, the so-called DMS summer paradox, thanks to the progressive dissociation between $Chl_{SAT}$ and DMS imposed by the two-step structure (Fig. 2) and the nonlinear relationships embodied in eq. 2 (Fig. 3). However, it fails to capture the high DMS/DMSPt ratios that occur in some regions between midsummer and early fall (Figs. 6 and 10), as discussed below.

| Page 19: [43] Deleted | Martí Galí Tàpias | 2018-05-08 12:39 |

Figs. 6-7 shows that, compared to $DMS_{SAT}$, the SD02 and VS07 algorithms produce higher DMS (and sometimes too high DMS) well into fall. This suggests that algorithms relying on MLD (SD02) or MLD combined with irradiance and water transparency (VS07) are better able to delay the annual DMS peak with respect to the summer solstice. E

Figs. 6-7 shows that, compared to $DMS_{SAT}$, the SD02 and VS07 algorithms produce higher DMS (and sometimes too high DMS) well into fall. This suggests that algorithms relying on MLD (SD02) or MLD combined with irradiance and water transparency (VS07) are better able to delay the annual DMS peak with respect to the summer solstice. E

| Page 19: [44] Deleted | Martí Galí Tàpias | 2018-05-14 15:39 |
|---|---|---|

At

| Page 19: [44] Deleted | Martí Galí Tàpias | 2018-05-14 15:39 |
|---|---|---|

At

| Page 19: [44] Deleted | Martí Galí Tàpias | 2018-05-14 15:39 |
|---|---|---|

At

| Page 19: [44] Deleted | Martí Galí Tàpias | 2018-05-14 15:39 |
|---|---|---|

At

| Page 19: [44] Deleted | Martí Galí Tàpias | 2018-05-14 15:39 |
|---|---|---|

At

| Page 19: [44] Deleted | Martí Galí Tàpias | 2018-05-14 15:39 |
|---|---|---|

At

| Page 19: [44] Deleted | Martí Galí Tàpias | 2018-05-14 15:39 |
|---|---|---|

At

| Page 19: [44] Deleted | Martí Galí Tàpias | 2018-05-14 15:39 |
|---|---|---|

At

| Page 20: [45] Deleted | Martí Galí Tàpias | 2018-05-15 16:12 |
|---|---|---|

| Page 20: [45] Deleted | Martí Galí Tàpias | 2018-05-15 16:12 |
|---|---|---|

| Page 20: [45] Deleted | Martí Galí Tàpias | 2018-05-15 16:12 |
|---|---|---|

| Page 20: [45] Deleted | Martí Galí Tàpias | 2018-05-15 16:12 |
|---|---|---|

| Page 20: [45] Deleted | Martí Galí Tàpias | 2018-05-15 16:12 |
|---|---|---|

| Page 20: [45] Deleted | Martí Galí Tàpias | 2018-05-15 16:12 |
|---|---|---|

| Page 20: [45] Deleted | Martí Galí Tàpias | 2018-05-15 16:12 |
|---|---|---|

| Page 20: [46] Deleted | Martí Galí Tàpias | 2018-05-15 19:13 |
|---|---|---|

fairly predictable

| Page 20: [46] Deleted | Martí Galí Tàpias | 2018-05-15 19:13 |

fairly predictable

| Page 20: [46] Deleted | Martí Galí Tàpias | 2018-05-15 19:13 |

fairly predictable

| Page 20: [46] Deleted | Martí Galí Tàpias | 2018-05-15 19:13 |

fairly predictable

| Page 20: [46] Deleted | Martí Galí Tàpias | 2018-05-15 19:13 |

fairly predictable

| Page 20: [46] Deleted | Martí Galí Tàpias | 2018-05-15 19:13 |

fairly predictable

| Page 20: [46] Deleted | Martí Galí Tàpias | 2018-05-15 19:13 |

fairly predictable

| Page 20: [46] Deleted | Martí Galí Tàpias | 2018-05-15 19:13 |

fairly predictable

| Page 20: [46] Deleted | Martí Galí Tàpias | 2018-05-15 19:13 |

fairly predictable

| Page 20: [46] Deleted | Martí Galí Tàpias | 2018-05-15 19:13 |

fairly predictable

| Page 20: [46] Deleted | Martí Galí Tàpias | 2018-05-15 19:13 |

fairly predictable

| Page 20: [46] Deleted | Martí Galí Tàpias | 2018-05-15 19:13 |

fairly predictable

| Page 20: [46] Deleted | Martí Galí Tàpias | 2018-05-15 19:13 |

fairly predictable

| Page 20: [47] Deleted | Martí Galí Tàpias | 2018-05-15 10:05 |

may be

| Page 20: [47] Deleted | Martí Galí Tàpias | 2018-05-15 10:05 |

may be

| Page 20: [47] Deleted | Martí Galí Tàpias | 2018-05-15 10:05 |

may be

| Page 20: [48] Deleted | Martí Galí Tàpias | 2018-05-15 10:03 |

like

| Page 20: [48] Deleted | Martí Galí Tàpias | 2018-05-15 10:03 |

like

| Page 20: [49] Deleted | Martí Galí Tàpias | 2018-05-08 15:33 |

T

| Page 21: [50] Moved to page 20 (Move #5) | Martí Galí Tàpias | 2018-05-08 15:34 |

Biotic interactions like microzooplankton grazing (Steiner et al., 2012) and bacterial metabolism (Levine et al., 2016) are indeed good candidates to explain strong deviations from the mean relationship between DMS, DMSPt and irradiance. However, they can hardly be included in empirical algorithms.

| Page 21: [51] Deleted | Martí Galí Tàpias | 2018-05-02 18:20 |

, which can account for biotic and abiotic DMS sources and sinks

| Page 21: [52] Deleted | Martí Galí Tàpias | 2018-05-15 10:47 |

as long as they can retrieve the relevant proxy variables

| Page 21: [53] Deleted | Martí Galí Tàpias | 2018-05-09 10:39 |

Yet, it cannot produce high DMS/DMSPt ratios in late summer, which suggests that irradiance cannot fully explain variability in DMS/DMSPt ratios in some regions. In the Antarctic Ocean, bias in satellite retrieved Chl causes a strong negative bias in $DMS_{SAT}$, which should be solved through regional tuning.

| Page 29: [54] Deleted | Martí Galí Tàpias | 2018-05-01 10:38 |

s.

| Page 29: [54] Deleted | Martí Galí Tàpias | 2018-05-01 10:38 |

s.

| Page 29: [54] Deleted | Martí Galí Tàpias | 2018-05-01 10:38 |

s.

| Page 29: [54] Deleted | Martí Galí Tàpias | 2018-05-01 10:38 |

s.

| Page 34: [55] Deleted | Martí Galí Tàpias | 2018-04-27 14:27 |

[Figure]

**Figure 6. DMS seasonal cycles by biomes**. The monthly means, medians, interquartile range and 5%-95% percentiles are shown for the in situ database, the L11 climatology, and remote sensing climatologies derived from the $DMS_{SAT}$, SD02 and VS07 algorithms. The temporal axis has been shifted by 6 months in the Southern hemisphere, i.e., July is the 1st month and June the 12th.

| Page 34: [56] Deleted | Martí Galí Tàpias | 2018-04-27 14:24 |
|---|---|---|

| Page 34: [56] Deleted | Martí Galí Tàpias | 2018-04-27 14:24 |
|---|---|---|

| Page 34: [56] Deleted | Martí Galí Tàpias | 2018-04-27 14:24 |
|---|---|---|

| Page 34: [56] Deleted | Martí Galí Tàpias | 2018-04-27 14:24 |
|---|---|---|

| Page 34: [56] Deleted | Martí Galí Tàpias | 2018-04-27 14:24 |
|---|---|---|

| Page 40: [57] Deleted | Martí Galí Tàpias | 2018-05-02 10:23 |
|---|---|---|

**Table 3: Global mean area-weighted DMS concentrations calculated with different algorithms**. Different $DMS_{SAT}$ results were obtained with alternative approaches for retrieving chlorophyll $a$ concentration ($Chl_{SAT}$) and the euphotic layer depth ($Zeu_{SAT}$) from satellite data. Calculations are based on 1°x1° gridded data; na: not applicable.

| DMS algorithm or data product | $Chl_{SAT}$ product | $Kd_{SAT}$ or $Zeu_{SAT}$ product | Area weighted DMS mean (nM) | | |
|---|---|---|---|---|---|
| | | | | | |
| L11 climatology(Lana et al., 2011) | na | na | | | |
| SD02(Simó and Dachs, 2002) | OC4-CI | na | | | |
| VS07(Vallina and Simó, 2007b) | na | Kd490 | | | |
| $DMS_{SAT}$eq. 2f(this study) | OC4-CI | Zeu = 4.6/Kd490 Zeu_Lee | | | |
| | GSM | Zeu_Lee | | | |
| | | | | | |